# GOLD: GLOBAL OVERVIEW TO LOCAL DETAIL IN EFFICIENT VISUAL GROUNDING FOR GUI AGENTS

## ABSTRACT

Graphical User Interface (GUI) agents powered by Vision-Language Models (VLMs) have recently emerged as a promising direction for multimodal automation. However, VLM-based GUI grounding incurs substantial computational overhead, making deployment on edge devices infeasible and leading to prohibitively high cloud serving costs. Prior attempts to reduce background or history vision tokens partially alleviate this issue, but either rely on sparsity in foreground elements or require extensive fine-tuning. In this work, we present GOLD, **G**lobal **O**verview to **L**ocal **D**etail for efficient GUI grounding that is tuning-free and robust across varying interface densities. GOLD operates in three stages. At the *Global Pruning Stage*, we downsample GUI screenshots and feed them into the VLM to identify relevant regions, thereby achieving efficient context reduction. In the *Local Refinement Stage*, only crops of detected regions are passed to the VLM at high resolution. To retain broader contexts, we aggregate the outputs of both stages to integrate both global and local information in *Global-Local Context Fusion Stage*. Experimental results show that GOLD reduces TFLOPs by 78%, while even improving accuracy by 0.7%p when it is integrated into the state-of-the-art GUI grounding method on the ScreenSpot-V2 benchmark. These findings highlight the efficiency of our global-to-local grounding framework.

## 1 INTRODUCTION

Vision-Language-Action (VLA) models enhance environmental perception and action generation by jointly reasoning over visual and textual modalities while grounding them in executable behaviors (Kim et al., 2024; Padalkar et al., 2023). This capability has driven progress in embodied domains such as robotics (Huang et al., 2024), autonomous driving (Arai et al., 2025), and gaming (Lifshitz et al., 2023). It is also increasingly promising in digital settings. In particular, extending VLAs to graphical user interfaces (GUIs) opens up opportunities for large-scale software automation and end-user assistance, where agents can navigate diverse applications and perform tasks on behalf of users. Unlike physical environments, however, GUIs require precise interpretation of small, fine-grained elements such as icons and buttons whose semantics are often critical for correct action (You et al., 2024). As a result, GUI-oriented VLAs must operate directly on high-resolution screenshots to capture these details faithfully. However, such inputs are computationally burdensome, increase inference latency, and drive up serving costs, ultimately hindering the scalability and deployment of VLA-based GUI agents in practice.

Recently, several efforts have sought to improve the efficiency of visual grounding of GUI Agents. For instance, FastV (Chen et al., 2024) introduces token dropping strategies at intermediate layers. However, this approach risks discarding important visual information before the model has fully understood the input. To selectively remove only meaningless tokens, ShowUI (Lin et al., 2025) filters out background tokens, but its effectiveness drops significantly when the foreground becomes dense. Similarly, SimpAgent (Chen et al., 2025) reduces redundant historical context tokens, improving efficiency in multi-step settings but offering limited benefits for single-image grounding. Collectively, these methods highlight the challenge of eliminating irrelevant information without sacrificing essential context for accurate grounding.

Motivated by this gap, we introduce *GOLD*, a Global Overview to Local Detail for efficient and accurate GUI grounding. Our approach operates in three stages: (i) In the *Global Pruning Stage*,

a downsampled screenshot is fed into the VLM, where attention maps are leveraged to identify relevant regions at minimal cost. This enables the model to efficiently filter out irrelevant areas while preserving salient interface components. (ii) In the *Local Refinement Stage*, only the candidate regions are re-examined at their original resolution. (iii) Finally, to maintain global context, we aggregate the attention maps from both stages, effectively integrating global, low-resolution context with local, high-resolution detail in *Global-Local Context Fusion Stage*.

We conduct extensive evaluations across tasks, baselines, and ablations to validate the effectiveness of GOLD. Experiments are performed on both GUI grounding and agent benchmarks, using ScreenSpot-V2 (Cheng et al., 2024) and Multimodal-Mind2Web (Zheng et al., 2024). Comparisons are made against state-of-the-art baselines for GUI grounding (OpenAI, 2025; 2024; Bai et al., 2025b;a; Wu et al., 2024; Hsieh et al., 2025; Xu et al., 2025; Lin et al., 2025; Wu et al., 2025), agent tasks (Zheng et al., 2024; Gou et al., 2025; Wu et al., 2025), and training-free visual compression methods (Chen et al., 2024; Zhang et al., 2024). Notably, when integrated with GUI-Actor (Wu et al., 2025), GOLD achieves new state-of-the-art results on ScreenSpot-V2, reducing TFLOPs by 78% while improving grounding accuracy by 0.7%p. These thorough evaluations confirm that GOLD's three-stage design and global–local context fusion not only cut computational overhead but also enhance accuracy, underscoring its robustness and efficiency in real-world GUI grounding applications.

We summarize our contributions as follows:

- We propose GOLD, a lightweight three-stage framework for GUI grounding that progressively prunes irrelevant regions, refines salient areas at their original resolution, and fuses the results for accurate prediction.
- We demonstrate that GOLD achieves substantial efficiency gains while improving accuracy, reducing TFLOPs by 78% and setting a new state of the art on ScreenSpot-V2.
- Through extensive evaluations and ablations, we show that the proposed global-local design is broadly applicable, providing a general strategy for achieving accurate grounding while remaining computationally efficient.

## 2 RELATED WORK

**GUI agents.** Recent advances in VLMs have spurred growing interest in their application to GUI-based task automation. Early studies primarily relied on outputting coordinate predictions for grounding instructions onto screen elements (Bai et al., 2025b; Hsieh et al., 2025; Qin et al., 2025; Wu et al., 2024; Gou et al., 2025). However, such approaches often suffer from weak alignment between spatial positions and natural language semantics (Wu et al., 2025). To address these challenges, recent studies have proposed alternative formulations. For instance, TAG (Xu et al., 2025) leverages the internal attention maps of VLMs rather than explicit coordinate tokens, enabling direct grounding in GUI contexts. Similarly, GUI-Actor (Wu et al., 2025) introduces attention-based action heads that significantly enhance grounding accuracy, achieving the state-of-the-art performance. Despite these advances, most of these methods still process entire screenshots at the original resolution, incurring substantial computational overhead. In contrast, our approach introduces a global-to-local grounding pipeline, dynamically detecting relevant crops on downsampled screenshots and performing grounding only within selected regions, thereby improving efficiency without sacrificing accuracy.

**Efficient GUI grounding.** Since VLM-based agents are computationally burdensome, researchers have started focusing on reducing the computational load of GUI grounding by pruning visual or historical context. FastV (Chen et al., 2024) proposes a vision-token dropping strategy applicable not only to GUI agents but also to general VLM inference. However, dropping tokens in intermediate layers risks discarding important features before the model has fully processed the image. ShowUI (Lin et al., 2025) and Iris (Ge et al., 2025) attempt to improve efficiency by removing background tokens deemed uninformative, though their gains diminish in cases where foreground elements are dense. SimpAgent (Chen et al., 2025), on the other hand, reduces the history tokens across multi-round interactions, but this approach primarily benefits long-horizon tasks and is not directly effective for single-image grounding. Moreover, both ShowUI and SimpAgent require additional training, which limits their applicability in dynamic or resource-constrained settings. In

contrast, our method adapts efficiently regardless of foreground element density and operates in a training-free manner. This eliminates the need for fine-tuning, significantly reducing computational overhead and deployment effort, while still ensuring robust grounding performance across diverse environments.

# 3 METHOD

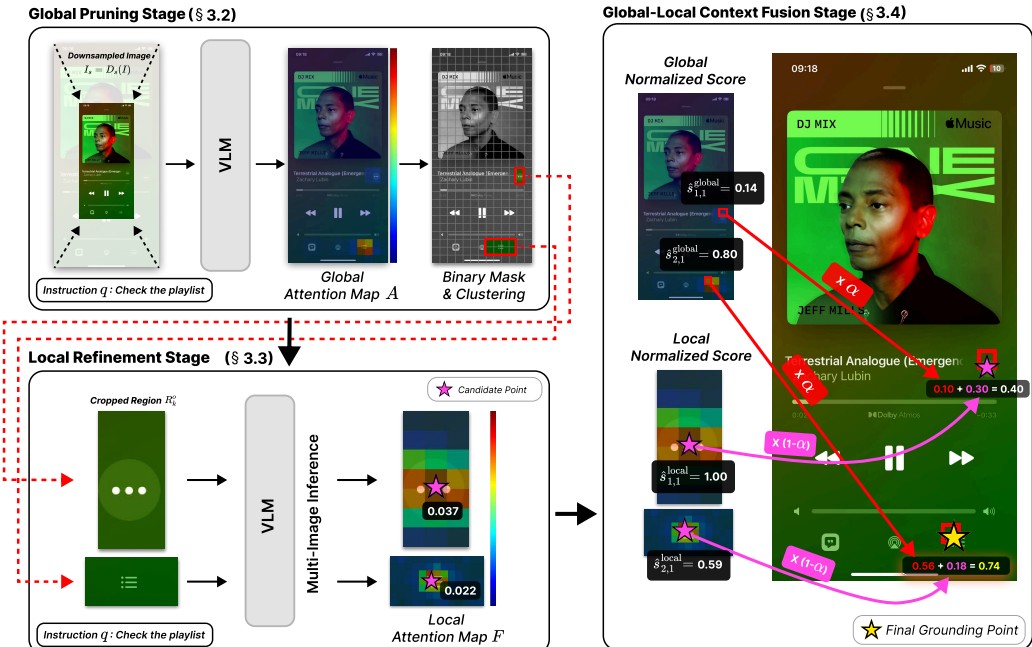

Figure 1: Illustration of the overall framework of GOLD.

## 3.1 OVERVIEW

We present *GOLD* (Global Overview to Local Detail) for efficient and accurate GUI grounding, a three-stage framework designed to make GUI grounding both efficient and accurate. The key challenge in this task is balancing global context with local precision: grounding requires awareness of the entire screen layout while also attending to small, fine-grained elements such as buttons, icons, or text. Processing the full-resolution image end-to-end with a VLM is computationally expensive, while using only downsampled inputs sacrifices detail. Our solution is a hierarchical pipeline that progressively narrows the search space and then reconciles global and local cues.

Figure 1 illustrates GOLD's overview. GOLD consists of three stages: (i) *Global Pruning Stage*, which downscales the input screen and identifies a small set of candidate regions using attention maps, filtering out irrelevant background at low cost; (ii) *Local Refinement Stage*, which restores these regions to their original resolution and re-encodes them to recover fine-grained details, producing precise candidate focal points; and (iii) *Global-Local Context Fusion Stage*, which integrates the coarse global context with the refined local evidence, ensuring the final prediction is both contextually aware and spatially precise.

This global-to-local design allows GOLD to achieve accurate grounding with significantly reduced computation, reusing signals already available at each stage rather than requiring redundant VLM forward passes.

## 3.2 GLOBAL PRUNING STAGE

The goal of the global pruning stage is to efficiently detect candidate regions from low-resolution inputs, thereby reducing computational overhead. Our intuition is that identifying broad regions of interest is simpler than pinpointing the exact action location; thus, high-resolution images are unnecessary at this stage. By lowering image resolution, we significantly cut computational cost while still preserving salient information. This lightweight pre-selection filters out irrelevant background and retains only promising regions, narrowing the search space for subsequent fine-grained grounding.

Specifically, we introduce a downsampling operator $D_s(\cdot)$, which rescales an image by a scaling factor $s \in (0, 1]$. We downsample the input screen $I$ to obtain $I_s = D_s(I)$ and then query a VLM to produce a global attention map:

$$\boldsymbol{A} = \text{Att}(I_s, q) \in \mathbb{R}^{h \times w},$$

where $h$ is the height and $w$ is the width of the screen. To localize areas of interest, we threshold the attention map at a relative level $\tau \in (0, 1]$ to obtain a binary mask:

$$\boldsymbol{M}_{ij} = \begin{cases} 1 & \text{if } \boldsymbol{A}_{ij} \geq \tau \cdot \max_{u,v} \boldsymbol{A}_{uv}, \\ 0 & \text{otherwise,} \end{cases} \qquad \tau \in (0, 1].$$

Each entry $\boldsymbol{M}_{ij}$ corresponds to a patch, the basic unit of processing in the VLM. A mask value of 1 indicates patches that are potentially relevant to the query.

Using the binary mask values, we form candidate regions by clustering patches with $\boldsymbol{M}_{ij} = 1$ if they are adjacent under 8-neighborhood connectivity (horizontal, vertical, or diagonal) following Xu et al. (2025). This clustering ensures that spatially contiguous patches are grouped together, preserving locality and structural coherence that are crucial for accurate grounding. Each connected cluster defines a candidate region $R_k$, which we score by summing its raw attention values:

$$\text{S}(R_k) = \sum_{(i,j) \in R_k} \boldsymbol{A}_{ij} \,.$$

To bound computation and memory, we retain up to $C$ top-scoring regions per image (default $C{=}3$). We discuss the effect of different $C$ values in Section 4.4. For each retained region, we compute an axis-aligned bounding box on the low-resolution grid, and map it back to the original coordinates. These cropped regions are then passed to the local refinement stage.

By operating on downsampled inputs, the global pruning stage effectively filters out irrelevant background while preserving promising regions, thereby reducing the cost of processing the full high-resolution image in the fine stage. While recent GUI grounding work has explored using attention maps for region selection (Wu et al. (2025); Zhang et al. (2025); Xu et al. (2025)), our contribution lies in making this process cost-efficient through an effective pruning stage followed by local refinement and fusion stages.

## 3.3 LOCAL REFINEMENT STAGE

The local refinement stage recovers details that may be lost at low resolution by re-examining the original-scale regions retained from the global pruning stage. While the global pruning stage efficiently prunes irrelevant background, its downsampled view inevitably blurs small but semantically important interface elements such as icons or text labels. The refinement stage addresses this by restoring resolution and enabling precise grounding within the filtered regions.

Let $\{R_k^o\}_{k=1}^C$ denote these regions, each cropped at the original image resolution without further resizing. We jointly encode all $C$ regions in a single multi-image prompt to the VLM, explicitly preserving their identities and original coordinate metadata for later fusion. This design avoids treating the regions as a single merged input, thereby reducing the loss of fine-grained details and enabling the model to perceive each region with greater fidelity. Importantly, this setup ensures that fine-grained perception benefits from both high-resolution detail and explicit spatial grounding, which is essential for distinguishing between visually similar interface elements.

For each region $R_k^o$ paired with query $q$, the VLM produces a fine-grained attention map:

$$\boldsymbol{F}_k = \text{Att}(R_k^o, q) \in \mathbb{R}^{h_k \times w_k},$$

where $h_k$ and $w_k$ denote the height and width of region $R_k^o$ at its original scale. From each map $\boldsymbol{F}_k$, we select the top $P$ locations with the highest scores (default $P=10$). For each candidate location $(x_{k,j}, y_{k,j})$ in region $k$, we record its score:

$$s_{k,j}^{\text{local}} = \boldsymbol{F}_k[y_{k,j}, x_{k,j}],$$

and define the normalized coordinates as

$$(\hat{x}_{k,j}, \hat{y}_{k,j}) = \left( \frac{x_{k,j}}{w_k}, \frac{y_{k,j}}{h_k} \right).$$

These normalized coordinates are then mapped back to the original image frame using the stored coordinates of the region, applying only scaling and translation without resampling.

The local refinement stage thus outputs a candidate set $\mathcal{P}$, which is passed unchanged to the fusion stage for score-level integration with the global coarse attention map $\boldsymbol{A}$:

$$\mathcal{P} = \bigcup_{k=1}^{C} \{(\hat{x}_{k,j}, \hat{y}_{k,j}, s_{k,j}) \mid j = 1, \ldots, P\}.$$

### 3.4 GLOBAL-LOCAL CONTEXT FUSION STAGE

While the global pruning stage efficiently captures the global structure and the local refinement stage recovers local details, neither is sufficient on its own. Global attention maps lack the resolution needed to distinguish visually similar interface elements, and local crops, although precise, lose the broader layout context once separated from the full image. In particular, because the global view preserves structural relationships, overall layout topology, and the underlying UI hierarchy, combining the two becomes especially important for GUI grounding. Reliable GUI grounding therefore requires integrating both global and local cues.

The fusion stage addresses this need by combining fine-level evidence from the candidate set $\mathcal{P}$ with coarse context preserved in the global attention map $\boldsymbol{A}$. Each candidate's normalized coordinate $(\hat{x}_{k,j}, \hat{y}_{k,j})$ is mapped back to the original image frame using stored crop metadata, yielding global coordinates $(x_{k,j}, y_{k,j})$. Because fusion directly reuses $\boldsymbol{A}$ and $\mathcal{P}$ computed upstream, it introduces no additional VLM forward passes, keeping the process computationally efficient.

The global score for each candidate is obtained by patch lookup on $\boldsymbol{A}$:

$$s_{k,j}^{\text{global}} = \boldsymbol{A}[y_{k,j}, x_{k,j}].$$

Both global and local scores are then normalized by their respective maxima to ensure comparability:

$$m_{\text{global}} = \max_{k,j} s_{k,j}^{\text{global}}, \qquad m_{\text{local}} = \max_{k,j} s_{k,j}^{\text{local}},$$

$$\hat{s}_{k,j}^{\text{global}} = \frac{s_{k,j}^{\text{global}}}{m_{\text{global}}}, \qquad \hat{s}_{k,j}^{\text{local}} = \frac{s_{k,j}^{\text{local}}}{m_{\text{local}}}.$$

The final fused score integrates both signals:

$$S_{k,j} = \alpha \, \hat{s}_{k,j}^{\text{global}} + (1 - \alpha) \, \hat{s}_{k,j}^{\text{local}}, \qquad \alpha \in [0, 1].$$

The final prediction is obtained by selecting the candidate with the maximum fused score:

$$(x^{\star}, y^{\star}) = \arg \max_{k,j} S_{k,j}.$$

By reconciling global layout information with fine-grained local cues, the fusion stage mitigates the weaknesses of each individual stage, ensuring grounding that is both contextually aware and spatially precise.

Importantly, the fusion stage is *extremely lightweight*: it performs only a few score lookups on precomputed coordinates without any additional VLM inference. Since all major computations are already handled in earlier stages, fusion adds virtually no overhead, completing in less than a millisecond on a CPU.

Table 1: Grounding accuracy (%) on ScreenSpot-V2. Bold type indicates those of the highest accuracy. Results by Operator and GPT-4o are from Wu et al. (2025).

| Model | Mobile | | Web | | Desktop | | Avg | |
|---|---|---|---|---|---|---|---|---|
| | Acc | TFLOPs | Acc | TFLOPs | Acc | TFLOPs | Acc | TFLOPs |
| Operator (OpenAI, 2025) | 44.9 | - | 88.9 | - | 86.1 | - | 70.8 | - |
| GPT-4o + OmniParser-v2 (OpenAI, 2024) | 86.7 | - | 74.8 | - | 79.1 | - | 80.6 | - |
| Qwen2-VL-2B (Bai et al., 2025b) | 9.6 | 68.0 | 5.9 | 74.9 | 8.7 | 28.3 | 9.3 | 60.0 |
| Qwen2.5-VL-3B Bai et al. (2025a) | 67.3 | 81.3 | 35.2 | 89.2 | 58.4 | 34.8 | 54.0 | 71.8 |
| OS-Atlas-4B (Wu et al., 2024) | 60.1 | 27.8 | 23.8 | 48.6 | 39.5 | 42.0 | 42.2 | 38.7 |
| ZonUI-3B (Hsieh et al., 2025) | 81.2 | 81.2 | 32.5 | 86.7 | 77.5 | 34.7 | 63.5 | 70.9 |
| TAG-8.5B (Xu et al., 2025) | 64.8 | 19.7 | 54.1 | 19.8 | 59.6 | 17.1 | 59.8 | 19.1 |
| ShowUI-2B (Lin et al., 2025) | 42.5 | 68.0 | 48.7 | 74.9 | 72.5 | 28.3 | 52.5 | 59.9 |
| ShowUI-2B TS (Lin et al., 2025) | 45.4 | 63.3 | 9.0 | 70.0 | 64.1 | 26.5 | 48.0 | 55.9 |
| GUI-Actor-2B (Wu et al., 2025) | 89.6 | 68.0 | 89.9 | 72.8 | 88.0 | 28.2 | 89.3 | 59.2 |
| GUI-Actor-3B (Wu et al., 2025) | 92.4 | 81.8 | 89.0 | 87.3 | **90.4** | 35.0 | 90.7 | 70.7 |
| GUI-Actor-2B + GOLD | 90.4 | **13.1** | 87.2 | **14.9** | 83.5 | **6.8** | 87.5 | **12.0** |
| GUI-Actor-3B + GOLD | **93.4** | 17.1 | **90.4** | 19.6 | 89.8 | 9.2 | **91.4** | 15.9 |

## 4 EXPERIMENTS

In this section, we evaluate the performance of GOLD on two widely used GUI grounding and navigation benchmarks: ScreenSpot-V2 (Wu et al., 2024) and Multimodal-Mind2Web (Zheng et al., 2024). To verify that our approach is not limited to a single model, we validate it not only on GUI-Actor but also on Qwen2.5-VL (Bai et al., 2025a), a strong foundation model frequently used for GUI grounding tasks. We use a scaling factor of $s = 0.5$ and a fusion ratio of $\alpha = 0.7$ for GOLD by default unless otherwise specified. We further perform ablation studies to analyze the effects of the scaling factor and the fusion ratio on grounding performance. All experiments are conducted on a single NVIDIA A100 80GB GPU.

### 4.1 GUI GROUNDING

**Dataset.** For GUI grounding evaluation, we use ScreenSpot-V2 dataset (Wu et al., 2024), a realistic benchmark specifically designed for visual GUI grounding. It contains 1,272 single-step instructions paired with corresponding bounding boxes over GUI elements, covering both text and widget/icon types. Compared to the original ScreenSpot (Cheng et al., 2024) dataset, ScreenSpot-V2 improves annotation quality by re-labeling 11.32% of samples that were previously incorrect, resulting in more reliable evaluation.

**Baselines.** We benchmark against several categories of models. For closed-source foundation models, we include OpenAI GPT-4o (OpenAI, 2024) and OpenAI Operator (OpenAI, 2025). For open-source foundation model, we evaluate Qwen2-VL (Bai et al., 2025b) and Qwen2.5-VL Bai et al. (2025a). We also compare against recent state-of-the-art GUI grounding systems, OS-Atlas (Wu et al., 2024) and ZonUI (Hsieh et al., 2025). For efficiency-oriented approaches, we consider ShowUI (Lin et al., 2025), where ShowUI-TS applies token selection to discard up to 50% of redundant background visual tokens. Finally, attention-based grounding approaches such as TAG (Xu et al., 2025) and GUI-Actor (Wu et al., 2025) are included as representative architectures that explicitly leverage structural attention mechanisms.

**Results.** As summarized in Table 1, state-of-the-art methods such as TAG and GUI-Actor achieve strong grounding performance on ScreenSpot-V2. Our GOLD further improves the results. In particular, GOLD not only surpasses the original GUI-Actor model in accuracy but also reduces computational cost by 78% in terms of TFLOPs. Compared to efficiency-oriented baselines such as ShowUI, our method attains higher accuracy while requiring significantly less computation, highlighting the effectiveness of adaptive token selection for efficient GUI grounding.

Table 2: Element accuracy (%) on Multimodal-Mind2Web. Results by Choice and SoM are from Zheng et al. (2024).

| Input | Planner | Grounding | Cross-Task | | Cross-Website | | Cross-Domain | | Avg | |
|---|---|---|---|---|---|---|---|---|---|---|
| | | | Acc | TFLOPs | Acc | TFLOPs | Acc | TFLOPs | Acc | TFLOPs |
| Image + Text | GPT-4 | Choice(Zheng et al., 2024) | 46.4 | - | 38.0 | - | 42.4 | - | 42.3 | - |
| | | SoM(Zheng et al., 2024) | 29.6 | - | 20.1 | - | 27.0 | - | 25.6 | - |
| Image | GPT-4 | UGround-V1-2B (Gou et al., 2025) | 46.4 | 234.1 | 46.2 | 232.5 | 46.5 | 233.9 | 46.4 | 233.7 |
| | | GUI-Actor-2B (Wu et al., 2025) | 60.8 | 21.7 | 59.7 | 21.7 | 52.8 | 21.8 | 55.6 | 21.8 |
| | | + GOLD | 50.8 | **5.7** | 53.8 | **5.9** | **53.1** | 5.8 | 52.7 | **5.8** |
| | | GUI-Actor-3B (Wu et al., 2025) | **52.5** | 27.7 | **54.7** | 27.7 | 52.8 | 27.7 | **53.0** | 27.7 |
| | | + GOLD | 52.0 | 8.3 | 53.0 | 8.6 | 52.5 | 8.4 | 52.5 | 8.4 |
| | GPT-4o | UGround-V1-2B (Gou et al., 2025) | 48.6 | 234.1 | 47.6 | 232.5 | 47.7 | 233.9 | 47.9 | 233.7 |
| | | GUI-Actor-2B (Wu et al., 2025) | 50.9 | 21.8 | 52.6 | 21.8 | 62.4 | 21.7 | 58.4 | 21.8 |
| | | + GOLD | 60.5 | **5.8** | **60.7** | **5.9** | 61.6 | **5.8** | 61.2 | **5.8** |
| | | GUI-Actor-3B (Wu et al., 2025) | **61.3** | 27.7 | **60.7** | 27.6 | **62.3** | 27.7 | **61.8** | 27.7 |
| | | + GOLD | 60.3 | 8.5 | **60.7** | 8.6 | 61.7 | 8.3 | 61.3 | 8.4 |

## 4.2 GUI AGENT EVALUATION

We conduct offline agent evaluation to measure grounding ability in realistic web tasks following Gou et al. (2025). In this setup, a planner model translates high-level user instructions into detailed grounding instructions, and our grounding model is responsible for selecting the correct GUI element. We use GPT-4 Turbo and GPT-4o models (OpenAI, 2024) as the planner and evaluate grounding performance conditioned on its outputs. We report element accuracy as our approach mainly focuses on the GUI grounding task.

**Dataset.** We use Multimodal-Mind2Web (Zheng et al., 2024), the multimodal extension of Mind2Web (Deng et al., 2023), for evaluation. The test split consists of 1,013 tasks across diverse websites. Each task includes a high-level instruction and a sequence of actions, with a screenshot before each action forming the golden trajectory. To enable reproducibility, we follow Gou et al. (2025) and leverage cached outputs from a GPT-based planner.

**Baselines.** We compare our approach to several grounding strategies commonly used in agent evaluation. Specifically, we consider planner-only methods, including Choice, where the planner selects from a filtered list of HTML elements, and SoM (Yang et al., 2023), where the input screenshot is overlaid with Set-of-Mark labels for element selection. These methods were included as baselines to examine planner performance without grounding support. We also evaluate against UGround (Gou et al., 2025), a powerful GUI grounding model tailored for this task, as well as GUI-Actor (Wu et al., 2025), the backbone model on which our approach is built.

**Results.** As summarized in Table 2, our framework significantly outperforms the UGround baseline in terms of efficiency, achieving 13%p higher element accuracy with substantially fewer TFLOPs. Compared to the backbone GUI-Actor model, our method attains nearly the same performance, while reducing TFLOPs by 69%. These findings demonstrate the efficiency of our framework in realistic agent settings, where both accuracy and computational cost are critical.

## 4.3 COMPARISON WITH TRAINING-FREE VISUAL COMPRESSION METHOD

In this section, we conduct a comparative evaluation against other training-free visual compression models. Particular emphasis is placed on benchmarking our approach against FastV (Chen et al., 2024) and SparseVLM (Zhang et al., 2024), both of which are widely regarded as strong and reliable visual compression methods. To further assess the practical effectiveness of our method, we apply it to the powerful foundation model Qwen2.5-VL as well as the state-of-the-art model ZonUI, and evaluate their performance on the ScreenSpot-V2 benchmark.

**Settings.** We evaluate the efficiency of our proposed attention-driven grounding method across different backbone models. Specifically, we focus on Qwen2.5-VL (Bai et al., 2025a), a widely used foundation model; GUI-Actor (Wu et al., 2025), a leading attention-driven GUI grounding backbone; and ZonUI (Hsieh et al., 2025), another recent state-of-the-art model that has shown strong

Table 3: Comparison between training-free visual compression method.

| Model | Mobile | | Web | | Desktop | | Avg | |
|---|---|---|---|---|---|---|---|---|
| | Acc | TFLOPs | Acc | TFLOPs | Acc | TFLOPs | Acc | TFLOPs |
| Qwen-2.5-VL 3B (Bai et al., 2025a) | 16.6 | 81.1 | 3.9 | 86.6 | 22.8 | 34.6 | 13.9 | 70.8 |
| + FastV (Chen et al., 2024) | 12.0 | 63.1 | 6.9 | 67.6 | 21.3 | 26.2 | 12.7 | 55.0 |
| **+ GOLD** | **40.1** | **18.8** | **23.8** | **20.4** | **40.4** | **11.6** | **34.6** | **17.5** |
| ZonUI-3B (Hsieh et al., 2025) | 24.8 | 81.1 | 9.4 | 86.6 | 27.5 | 34.6 | 20.2 | 70.8 |
| + FastV (Chen et al., 2024) | 37.3 | 63.1 | 13.5 | 67.6 | 29.9 | 26.2 | 27.2 | 55.0 |
| **+ GOLD** | **55.1** | **26.1** | **37.8** | **29.0** | **58.1** | **19.8** | **49.9** | **25.44** |
| GUI-Actor-2B (Wu et al., 2025) | 89.4 | 68.0 | **89.9** | 72.8 | **89.2** | 28.2 | **89.5** | 59.2 |
| + FastV (Chen et al., 2024) | 71.5 | 58.7 | 63.9 | 63.0 | 62.6 | 24.0 | 66.5 | 51.1 |
| + SparseVLM (Zhang et al., 2024) | 65.3 | 65.5 | 63.2 | 70.2 | 68.3 | 27.2 | 65.3 | 57.1 |
| **+ GOLD** | **90.6** | **12.6** | 87.6 | **14.4** | 84.4 | **6.43** | 88.0 | **12.6** |
| GUI-Actor-3B (Wu et al., 2025) | 92.4 | 81.8 | 89.0 | 87.2 | **90.4** | 35.0 | 90.7 | 70.7 |
| + FastV (Chen et al., 2024) | 68.6 | 63.4 | 68.2 | 68.0 | 74.3 | 26.5 | 70.0 | 55.3 |
| + SparseVLM (Zhang et al., 2024) | 59.9 | 74.9 | 55.4 | 80.0 | 69.2 | 31.9 | 60.8 | 65.4 |
| **+ GOLD** | **93.4** | **17.1** | **90.4** | **19.6** | 89.8 | **9.23** | **91.4** | **15.9** |

Table 4: Ablation study on each stage of GOLD.

| Model | Mobile | | Web | | Desktop | | Avg | |
|---|---|---|---|---|---|---|---|---|
| | Acc | TFLOPs | Acc | TFLOPs | Acc | TFLOPs | Acc | TFLOPs |
| Vanilla | 92.4 | 81.8 | 89.0 | 87.2 | 90.4 | 35.0 | 90.7 | 70.7 |
| + Global Stage | 92.4 | 14.3 | 83.8 | 17.4 | 76.7 | 7.5 | 85.3 | 13.6 |
| + Local Stage | 91.2 | 17.1 | 89.2 | 19.6 | 87.7 | 9.2 | 89.6 | 15.9 |
| + Fusion Stage | 93.4 | 17.1 | 90.4 | 19.6 | 89.8 | 9.2 | 91.4 | 15.9 |

performance in GUI grounding tasks. For Qwen2.5-VL and ZonUI, we introduce an attention-driven grounding mode. As a comparative baseline, we adopt FastV (Chen et al., 2024), a strong training-free token-reduction method that drops a fixed ratio of low-attention vision tokens at a designated layer. We also experiment with SparseVLM (Zhang et al., 2024), which prunes vision tokens based on important text tokens. Further implementation details are provided in Appendix A.4.

**Results.** Our results demonstrate consistent efficiency gains across both Qwen2.5-VL and GUI-Actor. FastV reduces TFLOPs by 15% when applied to Qwen2.5-VL, but this results in a drop in accuracy, as tokens are discarded before the model fully processes the screenshot context. In contrast, our method prunes unnecessary regions only after the model has already formed a global overview of the input. This design improves accuracy by 20%p, while achieving a substantial 77% reduction in TFLOPs. When applied to GUI-Actor-3B, our approach also achieves higher accuracy and lower TFLOPs than all other baselines. Overall, these results highlight the cost-effectiveness of our method compared to conventional token reduction strategies, underscoring its potential as a general efficiency enhancement technique for vision-language grounding.

## 4.4 Ablation Study

**Impact of each stage of GOLD.** To evaluate the contribution of each stage in GOLD, we perform an ablation study by incrementally adding them to the vanilla GUI-Actor-3B model on the ScreenSpot-V2 dataset. The results are summarized in Table 4. Incorporating the global pruning stage drastically reduces TFLOPs relative to the vanilla model, but applying it alone lowers accuracy due to the downsampling effect. Adding the local refinement stage recovers much of this loss, improving accuracy by about 4%p, although using it alone still underperforms the vanilla case because it lacks global context. Finally, introducing the global-local context fusion stage resolves this limitation, providing an additional accuracy gain of 1.81%p. Notably, the global-local context fusion stage introduces no extra TFLOPs, since it only performs score lookups on precomputed coordinates without additional VLM inference.

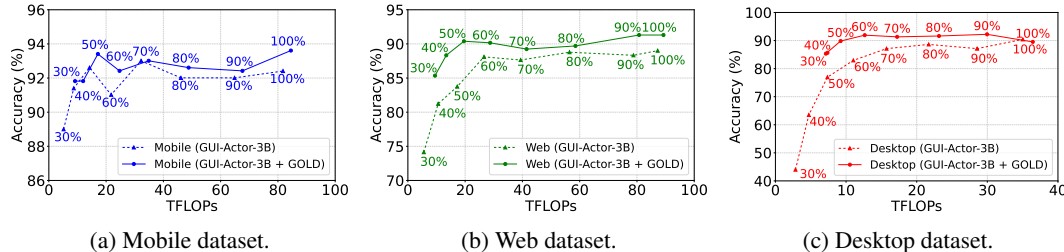

(a) Mobile dataset.  (b) Web dataset.  (c) Desktop dataset.

Figure 2: Grounding accuracy (%) versus TFLOPs on ScreenSpot-V2 as a function of the scaling factor $s$. The percentages next to each point indicate the value of $s$.

**Impact of scaling factor $s$.** We investigate the role of input resizing in our framework by varying the downsampling factor $s$ at the global pruning stage. Specifically, we compare two settings: (i) our method, where the screenshot image $I$ is downsampled by $s$ for global pruning and subsequently refined during the fine grounding stage and further enhanced at the fusion stage, and (ii) a baseline setting, where the same downsampling is applied directly to the vanilla model, GUI-Actor, for grounding without any refinement. All experiments are conducted on the ScreenSpot-V2 benchmark. As shown in Figure 2, the vanilla model exhibits a sharp performance drop, especially in web and desktop datasets, around the low scaling factor. This phenomenon can be attributed to the effect of resizing: fine-grained elements such as text and icons become increasingly blurry, thereby impairing grounding accuracy (a visual example is in Figure 4a). In contrast, our method avoids this degradation by selectively magnifying regions that demand fine-grained detail during the fine stage. Consequently, accuracy remains stable while computational cost does not increase substantially. These results demonstrate the cost-effectiveness of our framework, highlighting its ability to preserve fine-grained information under constrained computational budgets. Furthermore, by surpassing the baseline even in high-TFLOPs settings, our method proves that it not only ensures efficiency but also elevates grounding performance itself.

**Impact of fusion ratio $\alpha$.** In the global-local context fusion stage, $\alpha$ is defined as the fusion weight between the global attention map and the local attention map, determining how much emphasis is placed on local detail. Note that $\alpha = 0$ corresponds to using only the local attention map, while $\alpha = 1$ corresponds to using only the global attention map. Figure 3 shows that grounding accuracy reaches its minimum at $\alpha = 0$, indicating that relying solely on local detail without global overview leads to a loss of contextual information. Performance improves as $\alpha$ increases, achieving its peak at $\alpha = 0.7$ with 91.43%. However, accuracy decreases when $\alpha = 1$, reaching 90.49%, suggesting

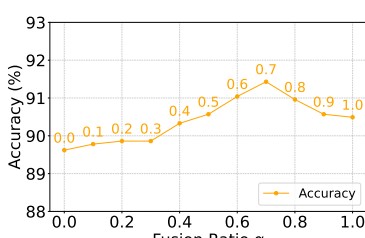

Figure 3: Impact of fusion ratio $\alpha$.

that discarding local detail is also suboptimal. Therefore, the results demonstrate that both global overview and local detail are essential, and optimal performance is achieved when they are balanced. By default, we use $\alpha = 0.7$ throughout the experiment.

**Impact of crop count $C$.** We investigate the effect of the crop count parameter $C$, which determines how many candidate regions from Global Pruning are forwarded to the subsequent high-resolution stages. We quantify this using Recall@$C$, defined as the probability that the ground-truth region remains within the Top-$C$ candidates after Global Pruning. As shown in Table 5, Recall@$C$ monotonically increases with larger $C$, but the final grounding accuracy saturates at $C = 3$ without further improvement for $C > 3$. This suggests that forwarding more than three crops does not contribute additional useful information for the downstream refinement.

Note that Recall@$C$ is different from the final accuracy: even if the ground-truth region survives the pruning stage, the model still needs Local Refinement and Fusion to precisely localize the point of interest within each candidate crop. Increasing $C$ only ensures that the correct region is retained, but it does not guarantee that the downstream stages will benefit from additional candidates.

Table 5: Recall@$C$ (Candidate Region Recall in Global Pruning).

| $C$ | Recall@$C$ | Final Acc |
|---|---|---|
| 1 | 95.36 | 91.27 |
| 2 | 97.41 | 91.35 |
| **3** | **97.80** | **91.43** |
| 4 | 98.03 | 91.43 |
| 5 | 98.19 | 91.43 |

Table 6: Fusion-selected region rank distribution (no pruning).

| Rank | Count | Percentage | Acc |
|---|---|---|---|
| 1 | 1257 | 98.82 | 90.78 |
| 2 | 12 | 0.94 | 50.00 |
| 3 | 2 | 0.16 | 100.00 |
| 4 | 0 | 0.00 | - |
| $\geq 5$ | 1 | 0.08 | 0.00 |

To further examine this phenomenon, we analyze which regions are actually selected during the final Fusion step when no pruning is applied. The distribution in Table 6 shows that 98.82% of all predictions rely on the top-1 region from Global Pruning, and 99.92% fall within the top-3 regions. Only a single case (0.08%) involves a region ranked 6, and this case results in an incorrect prediction. Thus, regions beyond the top-3 contribute negligibly to the final decision while increasing the computational cost of high-resolution inference.

## 5 CONCLUSION

In this paper, we propose GOLD (Global Overview to Local Detail) for efficient GUI grounding, a three-stage framework designed for efficient GUI grounding. GOLD first prunes irrelevant regions at the global pruning stage and subsequently refines predictions in the local refinement stage, enabling accurate decision-making in the global-local context fusion stage by integrating the outcomes of the previous stages. Experiments on the ScreenSpot-V2 and Multimodal-Mind2Web benchmarks demonstrate the effectiveness of our approach, showing that GOLD achieves high accuracy under constrained computational budgets. We believe our findings highlight a key principle for mobile AI: combining broad situational awareness with fine-grained precision yields robust performance without excessive computation. Looking ahead, we envision GOLD as a stepping stone toward AI agents that operate reliably in dynamic, resource-constrained environments while remaining efficient enough for everyday devices.

**Limitations and future work.** We observed that, in environments such as the Mobile dataset used for GUI grounding, the relatively large size of GUI elements allows the model to maintain high accuracy even when the scaling factor is substantially reduced. This finding suggests that the degree of resizing required may vary depending on the characteristics of GUI elements or other contextual factors. As future work, we aim to explore dynamically adjusting the resizing process based on these factors in order to achieve higher efficiency.

In addition, GOLD employs a cost-efficient Global–Local–Fusion architecture in which each high-resolution crop is processed independently before being combined through a lightweight fusion step. Although the Local Refinement Stage does not perform explicit cross-region self-attention, the Global Pruning Stage already provides context-aware region comparison by evaluating attention over the entire screen. This global stage effectively captures relationships between similar controls—such as two visually identical buttons in different panels—before cropping occurs. Nevertheless, incorporating lightweight mechanisms that further compare or relate regions could improve accuracy in cases involving highly symmetric or visually identical interfaces. Exploring this form of explicit cross-region reasoning, together with the associated accuracy-efficiency tradeoff, would be a promising direction for future research.

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

## LLM USAGE

An LLM was employed to improve sentence structure and ensure compliance with grammar rules.

## A APPENDIX

### A.1 PROMPTS USED FOR FINE-GRAINED FILTERING

---

**Prompts used for GOLD**

```
system prompt:
<Global Pruning Stage>
This is a resized screenshot of the whole GUI, scaled by
{resize_ratio}.
You are a GUI agent.
Given a screenshot of the current GUI and a human instruction, your
task is to locate the screen element that corresponds to the
instruction.
You should output a PyAutoGUI action that performs a click on the
correct position.
To indicate the click location, we will use some special tokens,
which is used to refer to a visual patch later.
For example, you can output:
pyautogui.click(<your_special_token_here>).

<Local Refinement Stage>
This is a list of {# of crops} cropped screenshots of the GUI, each
showing a part of the GUI.
You are a GUI agent.
Given a screenshot of the current GUI and a human instruction, your
task is to locate the screen element that corresponds to the
instruction.
You should output a PyAutoGUI action that performs a click on the
correct position.
To indicate the click location, we will use some special tokens,
which is used to refer to a visual patch later.
For example, you can output:
pyautogui.click(<your_special_token_here>).

user prompt:
```

---

### A.2 GROUNDING ACCURACY ON DIFFERENT MAX PIXEL

During evaluation, we further noticed a key sensitivity to resolution: the maximum pixel setting of the image processor significantly impacts both accuracy and computational cost (TFLOPs). For fair comparison, we fix the maximum pixel setting to 3,211,264, balancing accuracy and computational cost following previous work (Lu et al., 2025). However, as different models are pre-trained with varying max pixel settings, direct comparison may be unfair. To address this, we additionally report results under each model's default training resolution at Table 7. Specifically, ShowUI was trained with a maximum pixel resolution of 1,003,520, TAG with 1,806,336, and OS-Atlas with 1,204,224. Evaluation was conducted using the ScreenSpot-V2 benchmark.

Table 7: Grounding accuracy with different max pixel settings.

| Model | Avg | | Mobile | | Web | | Desktop | |
|---|---|---|---|---|---|---|---|---|
| | Acc | TFLOPs | Acc | TFLOPs | Acc | TFLOPs | Acc | TFLOPs |
| ShowUI-2B | 76.5 | 13.8 | 84.2 | 15.0 | 70.5 | 14.5 | 72.8 | 11.3 |
| TAG-8.5B | 59.8 | 19.0 | 64.9 | 19.7 | 54.1 | 19.8 | 59.6 | 17.1 |
| OS-Atlas-4B | 70.1 | 10.9 | 75.3 | 10.6 | 73.9 | **8.9** | 57.5 | 13.7 |
| GUI-Actor-3B | 89.3 | 18.3 | 92.8 | 19.7 | 85.8 | 19.1 | 88.6 | 15.1 |
| **+ GOLD** ($s = 0.3$) | 89.2 | **9.2** | 91.4 | **9.7** | 86.7 | 9.7 | 89.2 | **7.9** |
| **+ GOLD** ($s = 0.5$) | **91.9** | 16.2 | **92.8** | 17.6 | **91.5** | 20.0 | **91.0** | 9.4 |

## A.3 QUALITATIVE EXAMPLES

The following presents qualitative examples of the vanilla GUI-Actor and GOLD. Figure 4 illustrates a representative failure case where excessive screenshot downsampling causes the vanilla model to misidentify the target. In contrast, GOLD successfully corrects such cases by retaining the correct region during Global Pruning Stage and refining it through its Local Refinement Stage and Global-Local Context Fusion Stage. Figure 5 shows an example of results based on the device in ScreenSpot-V2. Figures 6 and 7 demonstrate cases where GOLD succeeds, thanks to its three-stage design. Figure 8 presents an ordinal-element example, showing that GOLD's Global Pruning Stage already captures the correct ordering and passes a reliable signal to the later stages.

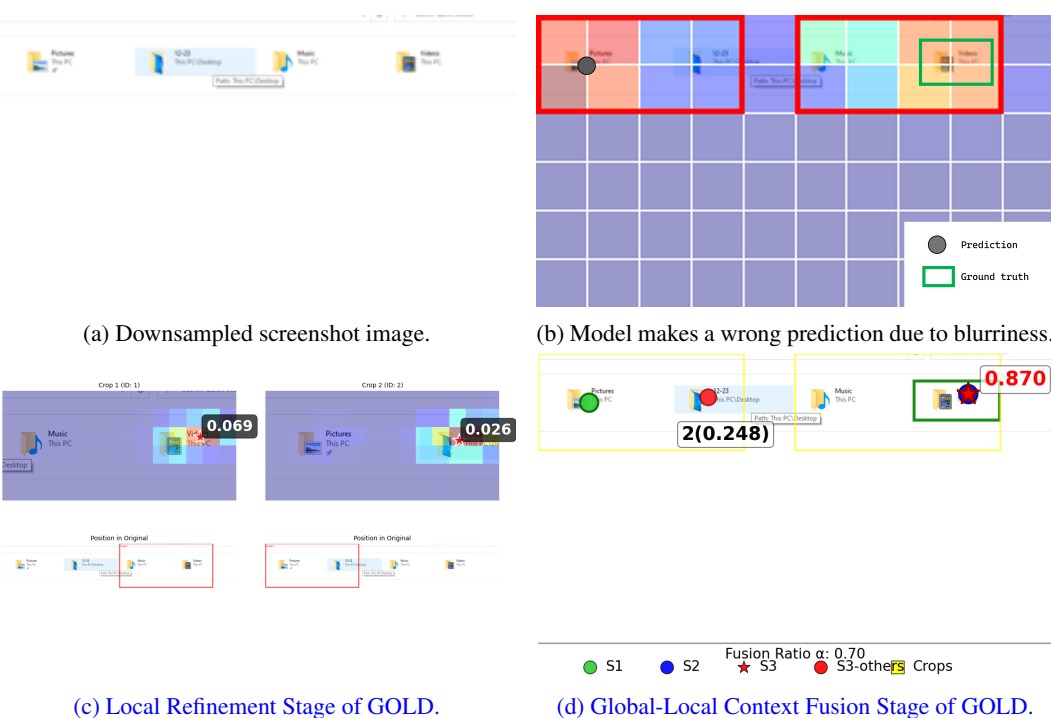

(a) Downsampled screenshot image.  (b) Model makes a wrong prediction due to blurriness.

(c) Local Refinement Stage of GOLD.  (d) Global-Local Context Fusion Stage of GOLD.

Figure 4: **Failure of vanilla grounding and recovery by GOLD.** Instruction: "Open videos folder." (a–b) Heavy downsampling introduces severe blurring and loss of fine-grained detail, causing the vanilla model—i.e., only the Global Pruning Stage of GOLD—to misidentify the target. (c) The correct region still appears among the pruned candidate crops, enabling the Local Refinement Stage to recover a clear high-resolution signal. (d) The Fusion Stage combines global region scores with the recovered local evidence to select the correct target.

## A.4 DETAILED IMPLEMENTATION DETAILS

For GUI-Actor, we use the attention map produced by the action head to guide grounding. For Qwen2.5-VL, we adopt an attention-driven grounding strategy that averages the attention maps from layers 20 to 31 to identify the highest-scoring patch. This choice is motivated by prior work (Luo et al., 2025), which shows that deeper layers provide stronger alignment between text prompts and the regions they attend to. We report a comparison of different layer ranges in Table 8, where the 20 to 31 range yields the best grounding accuracy.

For FastV, the procedure is similar to GUI-Actor but requires additional handling because Qwen2.5-VL prunes vision tokens at layer $K$. To determine a stable configuration for grounding, we sweep over pruning layers $K$ and pruning ratios $R$, as summarized in Table 9. Based on this analysis, we set $K = 3$ and $R = 0.7$, which provide a reasonable balance between grounding accuracy and computational cost. The pruned tokens are later reinjected by assigning zero scores in the final attention map to maintain structural consistency.

(a) ✓ (b) ✓ (c) ✓

Figure 5: **ScreenSpot-V2 examples of each TASK** (a) "start threadmill recording" in mobile, (b) "view all personal projects" in web, (c) "select python interpreter" in desktop. The green box indicates the ground truth, and the red dot represents model's prediction.

Table 8: Layer selection study for Qwen2.5-VL.

| # of layers | Layers | Acc |
|---|---|---|
| 36 | 0 - 35 | 19.2 |
| 12 | 0 - 11 | 1.6 |
| 12 | 4 - 15 | 5.5 |
| 12 | 8 - 19 | 6.9 |
| 12 | 12 - 23 | 30.0 |
| 12 | 16 - 27 | 31.4 |
| 12 | 20 - 31 | 32.2 |
| 12 | 24 - 35 | 19.9 |

Table 9: FastV pruning parameter selection.

| K | R | Acc | TFLOPs |
|---|---|---|---|
| 2 | 0.5 | 79.6 | 59.4 |
| | 0.7 | 75.2 | 55.1 |
| | 0.9 | 57.0 | 51.0 |
| 3 | 0.5 | 78.6 | 59.8 |
| | 0.7 | 72.0 | 55.5 |
| | 0.9 | 51.6 | 51.6 |
| 5 | 0.5 | 77.1 | 60.4 |
| | 0.7 | 70.0 | 56.4 |
| | 0.9 | 50.4 | 52.7 |

For SparseVLM, to remain consistent with the official implementation, the token recycling mechanism used a recycling ratio of $0.3$ to select top-ranked candidates from the deleted pool, and a cluster-center ratio $\theta$ of $0.1$ to determine the number of centroids for token compression. The sparsification scaling factor $\lambda$, which controls the layer-wise pruning intensity, was set to $0.02$.

## A.5 ADAPTIVE $\tau$ ESTIMATION

While we use a fixed threshold ($\tau = 0.12$) in the Global Pruning Stage to localize regions of interest, one may imagine adapting this threshold based on properties of the global attention distribution, by analyzing how the attention mass is concentrated or dispersed and selecting a data-driven cutoff that separates salient patches from background ones. Motivated by this, we additionally evaluate several adaptive rules, including Otsu thresholding (Otsu et al., 1975), a Pareto-front–style elbow heuristic (Satopaa et al., 2011), and a Shannon-entropy–based selection rule (Shannon, 1948). This experiment serves as an ablation to examine how different $\tau$-selection strategies influence region extraction quality.

**Implementation details.** For each method, we apply the adaptive rule directly to the raw global attention map $A$ (before normalization). Each method produces a scalar threshold $\tau$, which is then used to construct the binary mask $M_{ij} = \mathbf{1}[A_{ij} \geq \tau]$ exactly as in the main pipeline.

For the Pareto-front–style elbow heuristic, we flatten $A$, sort the values in descending order, compute adjacent differences, and select the point with the largest drop; the corresponding attention value becomes $\tau$. For Otsu thresholding, we lightly smooth $A$ with a Gaussian filter and apply the classical Otsu rule to the blurred map to obtain the threshold. For the entropy-based method, we normalize $A$ into a probability distribution, compute its Shannon entropy, and convert it into a threshold via $\tau = \exp(-H)$.

Table 10: Comparison of fixed and adaptive $\tau$ estimation methods across two scaling factors ($s$=0.3 and $s$=0.5).

| Model | Resize Ratio | Mobile | | Web | | Desktop | | Avg | |
|---|---|---|---|---|---|---|---|---|---|
| | | Acc | TFLOPs | Acc | TFLOPs | Acc | TFLOPs | Acc | TFLOPs |
| GUI-Actor-3B (Wu et al., 2025) | X | 92.4 | 81.8 | 89.0 | 87.2 | 90.4 | 35.0 | 90.7 | 70.7 |
| + Resizing | 0.3 | 89.0 | 5.2 | 74.1 | 5.6 | 44.0 | 2.8 | 72.1 | 4.7 |
| + GOLD ($\tau$ = 0.12) | 0.3 | **89.2** | 9.2 | **85.4** | 9.7 | **86.7** | 9.7 | **89.2** | 8.8 |
| + GOLD (**Pareto front** (Satopaa et al., 2011)) | 0.3 | 88.4 | 6.9 | 74.1 | 7.2 | 63.5 | 4.2 | 77.0 | 6.3 |
| + GOLD (**Entropy** (Shannon, 1948)) | 0.3 | 91.0 | 8.0 | 84.2 | 8.2 | 78.4 | 6.3 | 85.4 | 7.4 |
| + GOLD (**Otsu** (Otsu et al., 1975)) | 0.3 | 91.0 | 8.8 | 85.8 | 9.0 | 84.4 | 5.6 | 87.5 | 8.2 |
| + Resizing | 0.5 | 92.6 | 1.42 | 83.8 | 17.2 | 76.9 | 7.4 | 85.5 | 13.4 |
| + GOLD ($\tau$ = 0.12) | 0.5 | **93.4** | 17.1 | **90.4** | 19.6 | **89.8** | 9.2 | **91.4** | 15.9 |
| + GOLD (**Pareto front** (Satopaa et al., 2011)) | 0.5 | 91.2 | 15.7 | 83.8 | 18.7 | 81.4 | 8.6 | 86.1 | 14.9 |
| + GOLD (**Entropy** (Shannon, 1948)) | 0.5 | 93.2 | 16.4 | 88.8 | 19.2 | 88.0 | 9.2 | 90.3 | 15.4 |
| + GOLD (**Otsu** (Otsu et al., 1975)) | 0.5 | 92.4 | 16.9 | 89.7 | 19.5 | 89.5 | 8.9 | 91.0 | 15.8 |

Table 11: Latency Comparison.

| Model | Acc | Latency(s) |
|---|---|---|
| Qwen2.5-VL-3B Bai et al. (2025a) | 54.0 | 9.4 |
| ZonUI-3B (Hsieh et al., 2025) | 63.5 | 8.8 |
| ShowUI-2B TS (Lin et al., 2025) | 48.0 | 4.2 |
| OS-Atlas-4B (Wu et al., 2024) | 42.2 | 5.0 |
| GUI-Actor-2B (Wu et al., 2025) | 89.3 | 3.5 |
| GUI-Actor-2B (Wu et al., 2025) + **GOLD** | 87.5 | **0.7** |
| GUI-Actor-3B (Wu et al., 2025) | 90.7 | 4.4 |
| GUI-Actor-3B (Wu et al., 2025) + **GOLD** | **91.4** | 1.2 |

All subsequent stages (8-neighborhood clustering, region scoring, local refinement, and global–local fusion) remain unchanged. Thus, these adaptive rules serve as drop-in replacements for the fixed threshold in the Global Pruning Stage.

**Findings.** As shown in Table 10, the fixed threshold $\tau = 0.12$ achieves the highest accuracy across both $s = 0.3$ and $s = 0.5$. Among adaptive rules, Otsu performs closest to the fixed threshold, followed by entropy-based and Pareto-front thresholding. Although most adaptive strategies yield reasonable accuracy, they generally perform worse than the fixed $\tau$, suggesting that the simple relative threshold remains well aligned with our scenario. At the same time, these findings indicate the potential of fully automatic threshold selection. Adaptive $\tau$ mechanisms already show stable performance, and with further exploration—such as incorporating simple cues from the attention-map distribution (e.g., its sharpness or overall spread)—they may generalize better across diverse real-world GUI environments. In this work, the fixed threshold remains effective under the conditions studied, while adaptive thresholding presents a promising direction for future research.

## A.6 LATENCY COMPARISON

To demonstrate that the theoretical savings translate into practical speedups for real-world GUI agents, we also measured latency on an NVIDIA A100 80GB. As shown in Table 11, when applied to the GUI-Actor-3B model, GOLD achieves a 3.7× latency speedup while delivering the highest accuracy and the lowest TFLOPs.

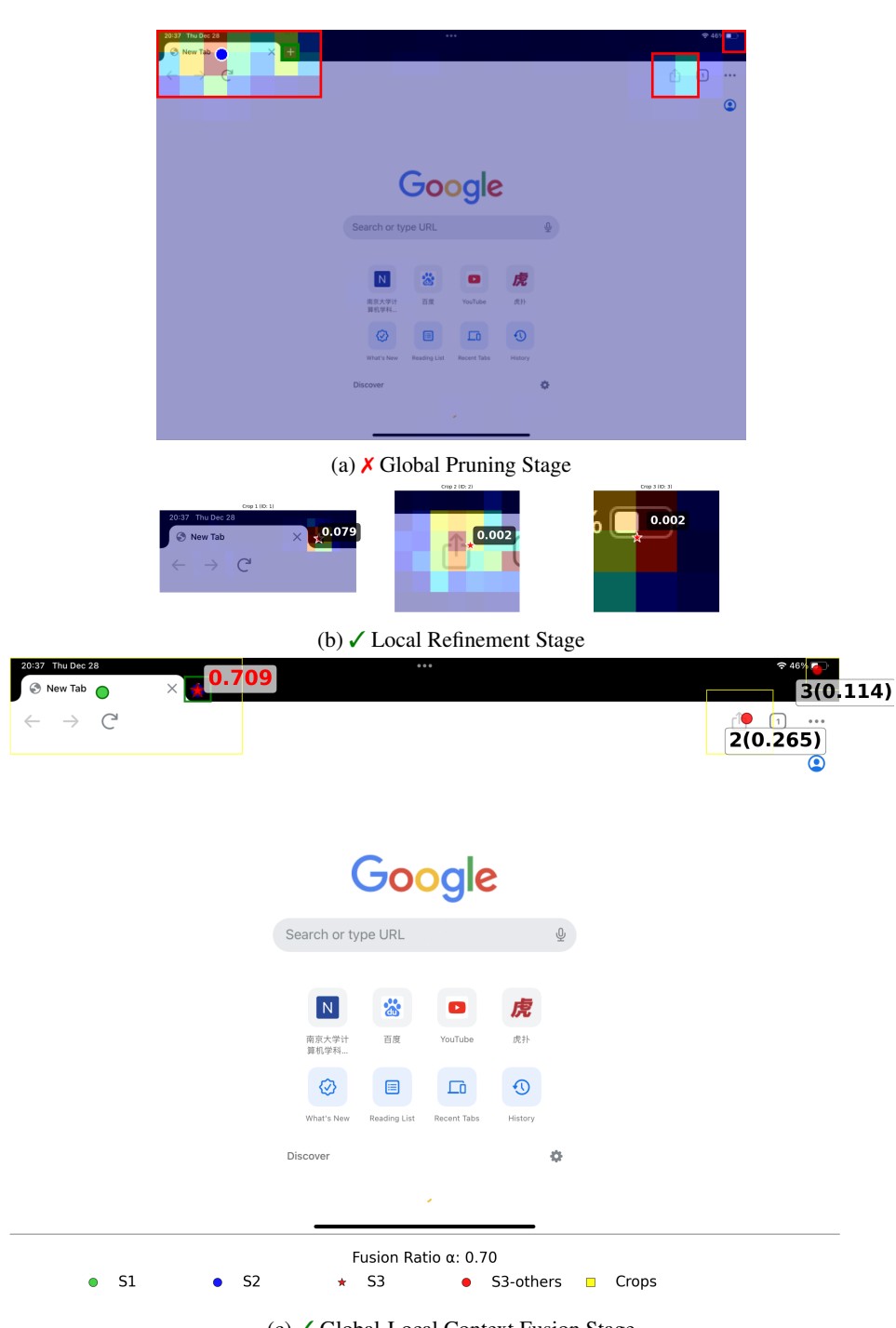

(a) ✗ Global Pruning Stage

(b) ✓ Local Refinement Stage

(c) ✓ Global-Local Context Fusion Stage

Figure 6: **Visualization of each stage.** Instruction: "add a new tab". In the Global Pruning Stage, the "new tab" button is too small to be directly identified, so the model focuses its attention on a broader region around the tab bar. Consequently, the nearby area is selected rather than the precise target. In the Local Refinement Stage, fine-grained details are captured, enabling the system to accurately locate and click the small content element — the "new tab" button itself. This demonstrates how the global-local pipeline progressively narrows down from coarse attention to precise action.

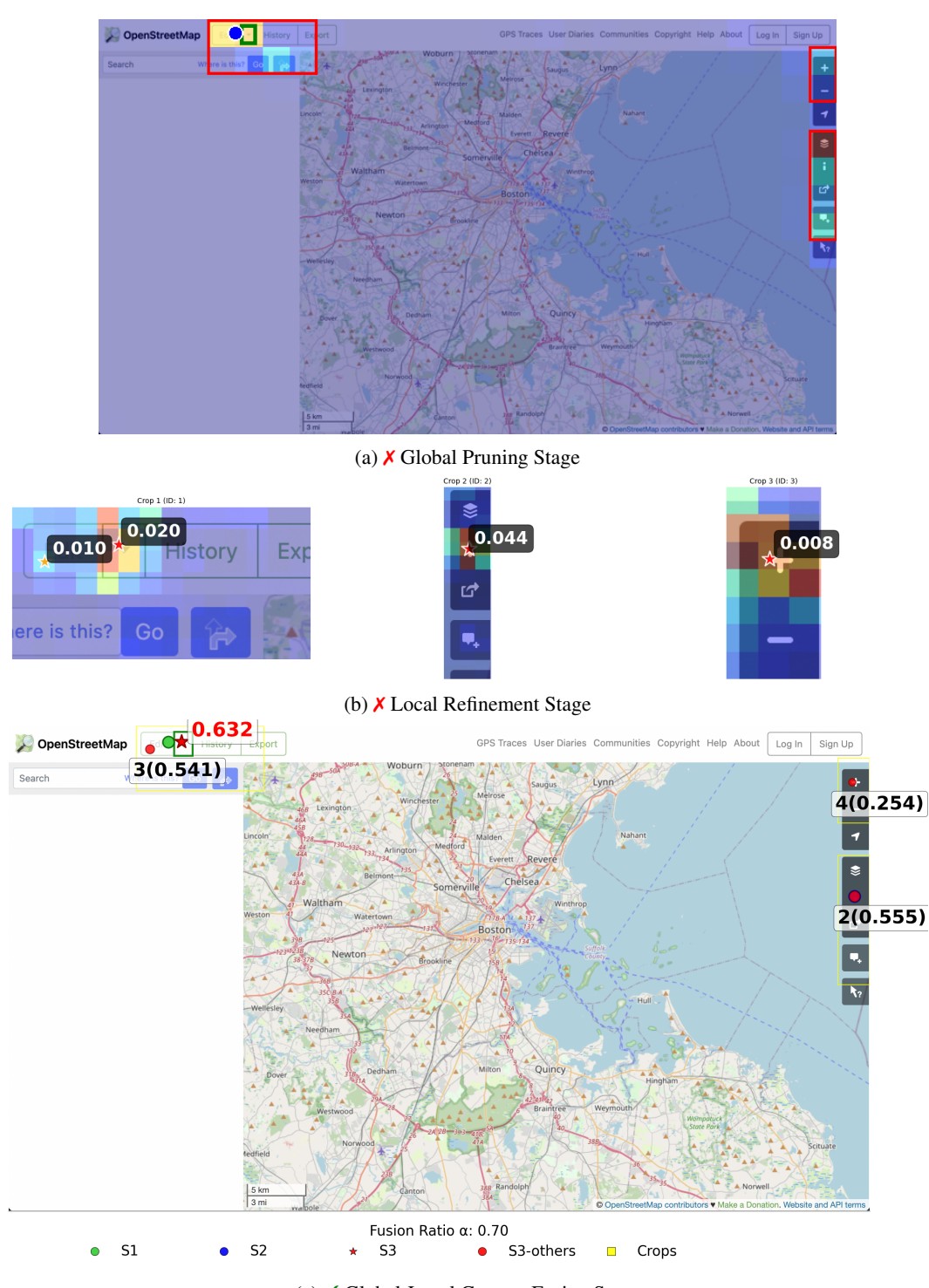

(a) ✗ Global Pruning Stage

(b) ✗ Local Refinement Stage

(c) ✓ Global-Local Context Fusion Stage

Figure 7: **Effect of Fusion Stage.** Instruction: "view more option of edit button". In the Global Pruning Stage, the model focuses on the correct general region along the top toolbar, but the low-resolution global map cannot resolve the small target precisely, leading to a near-miss. In the Local Refinement Stage, the crops are examined at high resolution but without global context; the model selects a salient element in a different crop altogether. After Fusion, the system combines the coarse global attention with the fine local candidates, re-ranking them to the intended control and correctly clicking the target.

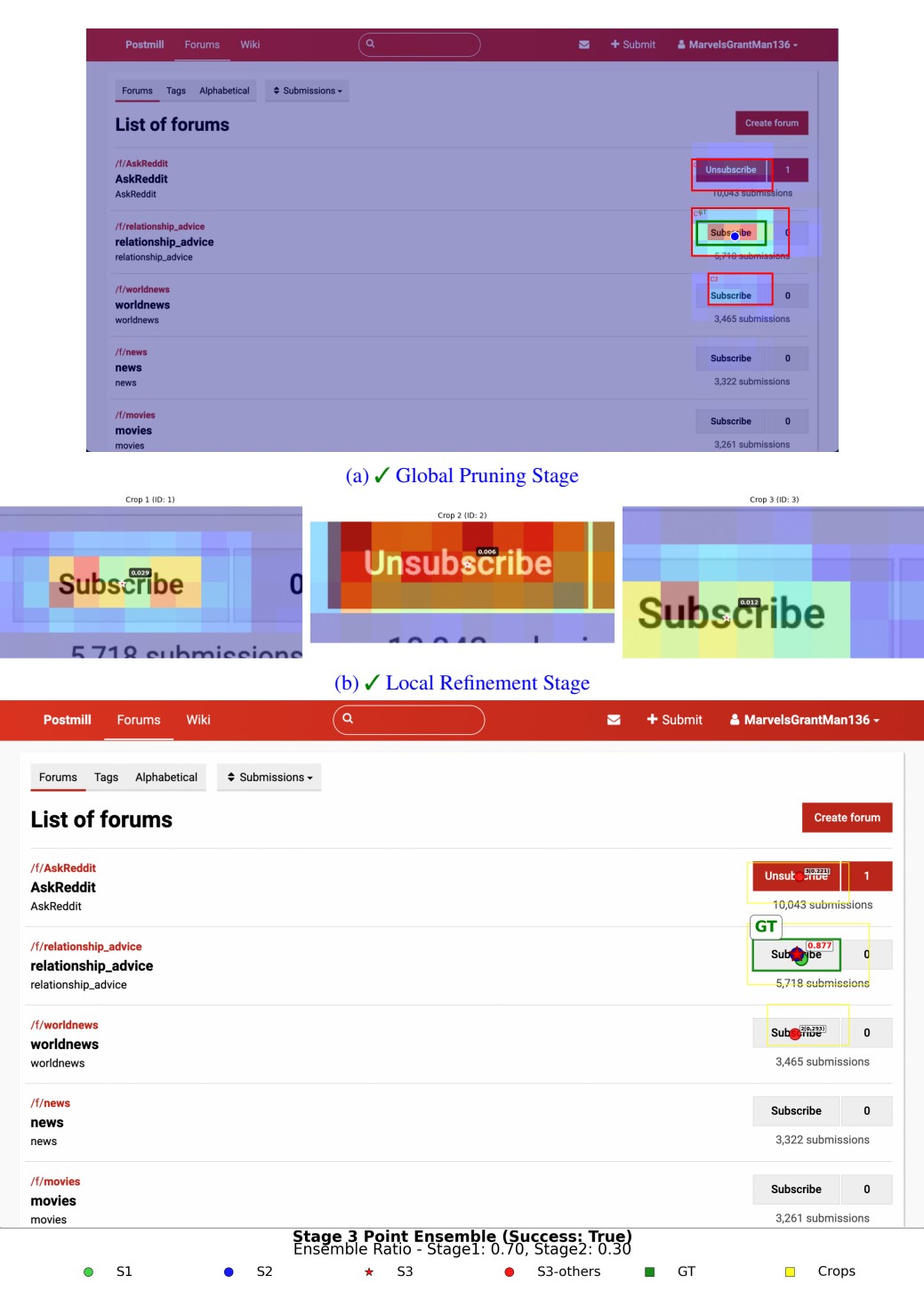

(a) ✓ Global Pruning Stage

(b) ✓ Local Refinement Stage

(c) ✓ Global-Local Context Fusion Stage

Figure 8: **Ordinal-element grounding.** Instruction: "subscribe the second book". In the Global Pruning Stage, the region corresponding to the second item receives a sufficiently strong attention response to remain within the Top-3 candidate crops. During Fusion, the system leverages the strong global signal to select this region over competing candidates, leading to the correct grounding.

