# OpenReview forum: "GOLD: Global Overview to Local Detail in Efficient Visual Grounding for GUI Agents"
_ICLR.cc/2026/Conference — Submitted to ICLR 2026_

### Official Review · Reviewer_FMFj · 2025-10-27

**Soundness:** 3
**Presentation:** 3
**Contribution:** 2
**Rating:** 6
**Confidence:** 3

**Summary:**

This paper presents **GOLD (Global Overview to Local Detail)**, a training-free framework for efficient visual grounding in GUI agents. GOLD operates in three stages: (1) a global pruning stage that downsamples the input screenshot and uses attention maps to identify candidate regions, (2) a local refinement stage that processes these regions at original resolution to recover fine-grained details, and (3) a global-local fusion stage that integrates both coarse and fine information to produce the final prediction. Evaluated on ScreenSpot-V2 and Multimodal-Mind2Web benchmarks, GOLD achieves a **78% reduction in TFLOPs** while improving grounding accuracy by 0.7%p when integrated with GUI-Actor. The method is plug-and-play, requiring no additional training, and demonstrates consistent improvements across multiple backbones and GUI environments (mobile, web, desktop).

**Strengths:**

**1. Practical and Training-Free**

The method requires no additional fine-tuning and can be seamlessly integrated into existing VLM-based GUI agents, making it highly practical for deployment in real-world systems.

**2. Clear and Lightweight Design**

The three-stage framework (global pruning, local refinement, global-local fusion) is simple, intuitive, and lightweight. The fusion stage adds virtually no computational overhead since it only performs score lookups on precomputed coordinates.

**3. Strong Empirical Results**

GOLD achieves up to 78% TFLOP reduction while slightly improving accuracy, even setting new state-of-the-art results on ScreenSpot-V2 when combined with GUI-Actor, with gains consistent across different backbones (GUI-Actor-2B/3B, Qwen2.5-VL) and platforms (Mobile, Web, Desktop). The paper also provides a comprehensive set of ablation studies to support the design choices.

**4. Great Presentation**

The paper is well-written with clear figures and organization, making the methodology easy to follow and supporting reproducibility and interpretability.

**Weaknesses:**

**1. Baseline Selection**

Most comparisons are made against other GUI grounding models (OS-Atlas, UGround, etc.) rather than efficiency-oriented methods. Since GOLD is positioned as an efficiency framework, it would be more informative to demonstrate its generalizability by applying it to a wider set of models reported in the paper and analyzing the resulting performance–efficiency trade-offs.

**2. Limited Comparison with Training-Free Acceleration for VLMs**

The evaluation only includes FastV as a training-free baseline, without clarifying whether its hyperparameters were carefully tuned for GUI grounding tasks. A broader comparison would strengthen the paper, particularly with recent token pruning approaches on VLMs (e.g., SparseVLM for adaptive sparsification) as well as grounding-specific optimizations such as GAP [arXiv:2506.21873], which addresses position ID misalignment after pruning, and FEATHER [arXiv:2412.13180], which proposes ensemble criteria for localization tasks. Including such baselines would help position GOLD more clearly within the growing space of efficient grounding methods.

**3. Novelty Concerns**

The core mechanism of attention-guided region selection followed by refinement bears some resemblance to the attention-based candidate selection already used in GUI-Actor. The paper would benefit from an explicit discussion of how GOLD differs.

More generally, the methodology follows standard coarse-to-fine grounding paradigms, which raises questions about (i) what is fundamentally new beyond hierarchical region selection and (ii) whether the framework could or should be validated on broader grounding tasks beyond GUIs.

**4. Practical Efficiency Metrics**

While reductions in TFLOPs are clearly reported, the multi-stage pipeline introduces additional forward passes. The paper does not provide latency or end-to-end runtime measurements (e.g., similar to UI-Agile’s reporting of wall-clock gains). Without such measurements on representative hardware, it is difficult to assess whether the theoretical FLOP savings translate into practical speedups for real-time GUI agents.

**Questions:**

1. Since GUI-Actor already employs attention-based action heads for grounding, could you explicitly clarify how GOLD's attention-driven region selection differs from GUI-Actor's existing mechanism? What specific advantages does the global-to-local pipeline provide beyond what GUI-Actor already implements?

2. The evaluation only compares with FastV as a training-free baseline. For FastV, were hyperparameters (pruning ratio, layer selection) carefully tuned for GUI grounding tasks? Could the authors include broader comparisons to other pruning/acceleration methods such as SparseVLM, GAP, or FEATHER?

3. While TFLOP reductions are reported, could the authors provide end-to-end latency/runtime measurements on representative hardware to show whether the theoretical savings translate into practical speedups for real-world GUI agents?

4. The methodology follows a general coarse-to-fine paradigm without apparent GUI-specific components. Could the authors clarify: (1) whether GOLD can be applied to standard visual grounding benchmarks like RefCOCO/RefCOCO+, (2) which design choices, if any, specifically exploit GUI properties such as structured layouts or small elements, and (3) whether evaluation on broader grounding tasks would strengthen the contribution, or if GOLD is intentionally tailored to GUIs?

5. Table 1, 2, 3 shows GOLD applied primarily to GUI-Actor and Qwen2.5-VL. Could you report results when applying GOLD to other models in the table (e.g., OS-Atlas, ZonUI, UGround)? This would better demonstrate the framework's generalizability as a plug-and-play efficiency enhancement.

---

> ### Author Response · Authors · 2025-11-21
> **Author Response [1/3]**
>
> We sincerely appreciate your time and effort in providing these valuable comments. We address your concerns and questions below, and we have also uploaded an `updated manuscript` that incorporates our rebuttal.
>
> ---
>
> **[Question 1 (Weakness 3)]**
> > Since GUI-Actor already employs attention-based action heads for grounding, could you explicitly clarify how GOLD's attention-driven region selection differs from GUI-Actor's existing mechanism? What specific advantages does the global-to-local pipeline provide beyond what GUI-Actor already implements?
>
>
> **[Response]**
>
> Our method and contributions are distinct from GUI-Actor. Indeed, using attention maps for region selection has been widely adopted in recent GUI grounding studies ([r1], [r2], [r3]). **Our contribution lies in making this process cost-efficient** through an effective pruning stage, followed by local refinement and fusion stages.
>
> Specifically, while both GOLD and GUI-Actor employ attention for region selection, GUI-Actor processes the entire high-resolution screenshot, which inherently incurs heavy computation. In contrast, **GOLD first applies attention to an aggressively downsampled global view to prune away irrelevant regions at very low cost**, and then performs high-resolution inference only within these selected areas, followed by a global-local fusion stage that restores contextual fidelity. This three-step pipeline (global-local-fusion) achieves large TFLOP reductions while maintaining accurate grounding (**reduces TFLOPs of GUI-Actor by 78%, while even improving accuracy by 0.7%p**), and this is precisely what sets it apart from GUI-Actor. We have also revised the text in  `Section 3.2` to clarify this:
> > While recent GUI grounding work has explored using attention maps for region selection (Wu et al. (2025); Zhang et al. (2025); Xu et al. (2025)), our contribution lies in making this process cost-efficient through an effective pruning stage followed by local refinement and fusion stages.
>
>
> [r1] Wu, Qianhui, et al. "GUI-Actor: Coordinate-Free Visual Grounding for GUI Agents." arXiv preprint arXiv:2506.03143 (2025).
>
> [r2] Zhang, Jiarui, et al. "MLLMs Know Where to Look: Training-free Perception of Small Visual Details with Multimodal LLMs." ICLR 2025.
>
> [r3] Xu, Hai-Ming, et al. "Attention-driven gui grounding: Leveraging pretrained multimodal large language models without fine-tuning." AAAI 2025.

---

> ### Author Response · Authors · 2025-11-21
> **Author Response [2/3]**
>
> **[Question 2 (Weakness 2)]**
> >The evaluation only compares with FastV as a training-free baseline. For FastV, were hyperparameters (pruning ratio, layer selection) carefully tuned for GUI grounding tasks? Could the authors include broader comparisons to other pruning/acceleration methods such as SparseVLM, GAP, or FEATHER?
>
>
> **[Response]**
> Thank you for your suggestion. We have expanded the hyperparameter sweep for FastV by referencing the hyperparameter ranges described in the FastV paper [r4]. The result is shown below.
>
> | K | R | Acc | TFLOPs |
> | --- | --- | --- | --- |
> | 2 | 0.5 | 79.6 | 59.4 |
> | 2 | 0.7 | 75.2 | 55.1 |
> | 2 | 0.9 | 57.0 | 51.0 |
> | 3 | 0.5 | 78.6 | 59.8 |
> | 3 | 0.7 | 72.0 | 55.5 |
> | 3 | 0.9 | 51.6 | 51.6 |
> | 5 | 0.5 | 77.1 | 60.4 |
> | 5 | 0.7 | 70.0 | 56.4 |
> | 5 | 0.9 | 50.4 | 52.7 |
>
> We found that GOLD (Acc: 91.4%; TFLOPS: 15.9) still substantially outperforms FastV even under this broader and more faithful tuning, achieving both the lowest computational cost and the highest accuracy. These results further confirm that the performance gap is not due to suboptimal FastV settings. We have included this result in `Table 9`.
>
>
> In addition, we have compared with SparseVLM [r5] as shown below.
>
> | Model                         | Mobile Acc | Mobile TFLOPs | Web Acc | Web TFLOPs | Desktop Acc | Desktop TFLOPs | Avg Acc | Avg TFLOPs |
> |------------------------------|------------|----------------|---------|-------------|--------------|-----------------|---------|-------------|
> | GUI-Actor-3B                 | 92.4       | 81.8           | 89.0    | 87.2        | **90.4**     | 35.0           | 90.7    | 70.7        |
> | + FastV                      | 68.6       | 63.4           | 68.2    | 68.0        | 74.3         | 26.5           | 70.0    | 55.3        |
> | + SparseVLM                  | 59.9       | 74.9           | 55.4    | 80.0        | 69.2         | 31.9           | 60.8    | 65.4        |
> | **+ GOLD**                 | **93.4**   | **17.1**       | **90.4**| **19.6**    | 89.8         | **9.23**       | **91.4**| **15.9**    |
>
> Compared to the vanilla baseline (GUI-Actor 3B), SparseVLM reduces TFLOPs by 8%, whereas GOLD achieves a much larger 78% reduction while also delivering higher accuracy, demonstrating its superior efficiency. We have incorporated this result in `Table 3`.
>
>
> [r4] Chen, Liang, et al. "An image is worth 1/2 tokens after layer 2: Plug-and-play inference acceleration for large vision-language models."ECCV 2024.
>
> [r5] Zhang, Yuan, et al. "Sparsevlm: Visual token sparsification for efficient vision-language model inference." ICML 2025.
>
>
> ---
>
> **[Question 3 (Weakness 4)]**
> >While TFLOP reductions are reported, could the authors provide end-to-end latency/runtime measurements on representative hardware to show whether the theoretical savings translate into practical speedups for real-world GUI agents?
>
>
> **[Response]**
> Thank you for your suggestion. We additionally measured the end-to-end latency on the same execution environment using an NVIDIA A100 80GB GPU:
>
> | Model | Acc | Latency(s) |
> | --- | --- | --- |
> | Qwen2.5-VL-3B | 54.0 | 9.4 |
> | ZonUI-3B | 63.5 | 8.8 |
> | ShowUI-2B TS | 48.0 | 4.2 |
> | OS-Atlas-4B | 42.2 | 5.0 |
> | GUI-Actor-2B | 89.3 | 3.5 |
> | GUI-Actor-2B + GOLD | 87.5 | **0.7** |
> | GUI-Actor-3B | 90.7 | 4.4 |
> | GUI-Actor-3B + GOLD | **91.4** | 1.2 |
>
> As shown, GOLD reduces the average end-to-end runtime from 4.4s to 1.2s (3.7 times faster) on GUI-Actor 3B, showing that the theoretical FLOP savings indeed translate into practical speedups and yield the lowest latency among all baselines. We have included this result in `Appendix A.6` and `Table 11`.

---

> ### Author Response · Authors · 2025-11-21
> **Author Response [3/3]**
>
> **[Question 4 (Weakness 3)]**
> >The methodology follows a general coarse-to-fine paradigm without apparent GUI-specific components. Could the authors clarify: (1) whether GOLD can be applied to standard visual grounding benchmarks like RefCOCO/RefCOCO+, (2) which design choices, if any, specifically exploit GUI properties such as structured layouts or small elements, and (3) whether evaluation on broader grounding tasks would strengthen the contribution, or if GOLD is intentionally tailored to GUIs?
>
>
> **[Response]**
>
> **GOLD is intentionally designed for efficient GUI grounding, and its three-step pipeline—global pruning, local refinement, and fusion—reflects this goal**. While the overall structure resembles a coarse-to-fine paradigm, several design choices are tailored specifically to GUI properties.
>
> First, the global pruning stage leverages a low-resolution global view that **preserves structural relations, layout topology, and UI hierarchy among elements**. This is particularly important for GUI grounding, where function often depends more on hierarchical context than on raw visual similarity. For instance, as illustrated in Figure 8, an instruction like “subscribe to the second book” may correspond to two visually identical “Subscribe” buttons belonging to different forum sections. Their appearance is indistinguishable, but their UI hierarchy—captured through the global attention pattern—is different. GOLD explicitly retains and uses this global hierarchical cue, whereas general coarse-to-fine approaches (such as [r6]) typically discard the global view after narrowing the search space and therefore risk losing such contextual signals. Furthermore, GOLD’s fusion stage integrates hierarchical global information with fine-grained local signals, enabling accurate grounding while maintaining low computational cost. **This makes GOLD well aligned with the characteristics of GUI tasks, where elements are dense, small, and structurally constrained**.
>
> Regarding applicability outside GUI settings: in principle, the coarse-to-fine decomposition could be applied to standard visual grounding datasets such as RefCOCO/RefCOCO+. However, these datasets do not require the type of hierarchical UI reasoning that GOLD is specifically designed to exploit, and therefore may not fully reflect the strengths of our approach. For this reason, we believe that evaluation on GUI-focused benchmarks—where structural layout and inter-element relations play a central role—is the most appropriate way to highlight GOLD’s contributions.
>
> We have revised `Section 3.4` to clarify these points and better articulate why GOLD is intentionally tailored to GUI grounding rather than general-purpose grounding tasks.
>
> [r6]  Luo, Junwei, et al. "When Large Vision-Language Model Meets Large Remote Sensing Imagery: Coarse-to-Fine Text-Guided Token Pruning." ICCV 2025.
>
>
>
>
>
> ---
>
> **[Question 5 (Weakness 1)]**
> >Table 1, 2, 3 shows GOLD applied primarily to GUI-Actor and Qwen2.5-VL. Could you report results when applying GOLD to other models in the table (e.g., OS-Atlas, ZonUI, UGround)? This would better demonstrate the framework's generalizability as a plug-and-play efficiency enhancement.
>
> **[Response]**
>
> Thank you for the suggestion. We additionally compared with ZonUI, and the result is shown below:
>
> | Model                         | Mobile Acc | Mobile TFLOPs | Web Acc | Web TFLOPs | Desktop Acc | Desktop TFLOPs | Avg Acc | Avg TFLOPs |
> |------------------------------|------------|----------------|---------|-------------|--------------|-----------------|---------|-------------|
> | GUI-Actor-3B                 | 92.4       | 81.8           | 89.0    | 87.2        | **90.4**     | 35.0           | 90.7    | 70.7        |
> | + FastV                      | 68.6       | 63.4           | 68.2    | 68.0        | 74.3         | 26.5           | 70.0    | 55.3        |
> | **+ GOLD**                 | **93.4**   | **17.1**       | **90.4**| **19.6**    | 89.8         | **9.23**       | **91.4**| **15.9**    |
> | ZonUI-3B | 24.8  | 81.1  | 9.4   | 86.6  | 27.5  | 34.6  | 20.2  | 70.8   |
> | + FastV | 37.3  | 63.1  | 13.5  | 67.6  | 29.9  | 26.2  | 27.2  | 55.0   |
> | **+ GOLD**| 55.1 | 26.1 | 37.8 | 29.0 | 58.1 | 19.8 | 49.9 | 25.44 |
>
> Compared to the original ZonUI without GOLD, the ZonUI + GOLD results show clear accuracy gains while maintaining consistently lower TFLOPs. We have incorporated the updated results into `Table 3` and believe that these additional comparisons further illustrate GOLD’s generalizability.

---

### Official Review · Reviewer_f7o6 · 2025-10-31

**Soundness:** 2
**Presentation:** 3
**Contribution:** 2
**Rating:** 4
**Confidence:** 2

**Summary:**

This paper proposes GOLD, a "global overview - local refinement" three-step method, which is used to significantly reduce the computational cost of VLM in GUI grounding without the need for fine-tuning. By first screening candidate regions at low resolution, then cropping the regions of the original image for high-resolution re-encoding, and finally fusing the global-local attention scores, a 78% TFLOPs reduction is achieved, while the accuracy is improved by 0.7% on benchmarks such as ScreenSpot-V2. The method is simple, training-independent, and suitable for deployment on mobile devices and cloud sides.

**Strengths:**

1.Training is irrelevant, plug-and-play, friendly to existing VLMs

2.The three-step pipeline has a clear structure, balancing both global context and local details

3.It achieves significant acceleration and improvement in accuracy simultaneously on multiple public benchmarks, with thorough experiments

4.It remains robust in foreground-intensive scenarios, outperforming previous methods such as background pruning or token dropping

**Weaknesses:**

1. Currently, τ ∈ (0, 1] is a fixed relative threshold, and C = 3 is an empirical upper limit. The optimal values under different GUI densities, resolutions, or screen sizes have not been explored. Figure 2 shows that when s = 0.3, the desktop accuracy has decreased by 4%, indicating that the attention peaks at low resolutions are easily "overwhelmed" by large-sized controls, and the 8-neighborhood clustering easily merges controls with small spacings into the same area. The paper does not provide adaptive estimation of τ (such as Otsu, Pareto front) or online feedback strategies, resulting in the need for manual adjustment of hyperparameters, thereby weakening the "no-training" advantage.

2.When the instructions involve "selecting three checkboxes simultaneously" or "clicking the fifth item in the menu", the model needs to focus on ≥3 areas. GOLD first performs hard truncation and then fusion. Once the target control element does not enter the Top-C, it cannot be restored. The experiment only reports the success rate of single-click, does not calculate the Recall@C curve, and does not provide an ablation study for C > 3. It is difficult to prove that C = 3 is sufficient for complex tasks.

3.The author assumes that "the low-resolution attention peak is sufficient to indicate the high-resolution target location", but when image compression causes text blurring and icons to have jagged edges, VLM attention often disperses to adjacent text or blank areas. Figure 4 is an example that fails due to blurring, but the paper only qualitatively presents this, without quantifying the failure rate, nor providing secondary verification (such as edge gradients, OCR boxes) to correct the candidate regions. Once the attention map is unreliable, the three-step pipeline of GOLD will go completely wrong, lacking rollback or self-check mechanisms.

4.To reduce the number of forward passes in VLM, the authors combined C high-resolution cropped images into multiple images for input at once. Although this approach is efficient, the self-attention mechanism of VLM only computes within a single image, and there is no interaction between regions. For instructions like "Click the 'Save' button in Panel A instead of the 'Save' button in Panel B", which require comparing two similar controls, the model cannot utilize cross-region attention for disambiguation. It can only rely on subsequent score fusion, but the fusion stage lacks an inter-region comparison mechanism.

**Questions:**

See Weaknesses

---

> ### Author Response · Authors · 2025-11-21
> **Author Response [1/4]**
>
> We sincerely appreciate your time and effort in providing these valuable comments. We address your concerns and questions below, and we have also uploaded an `updated manuscript` that incorporates our rebuttal.
>
> ---
>
> **[Weakness 1]**
> >Currently, τ ∈ (0, 1] is a fixed relative threshold, and C = 3 is an empirical upper limit. The optimal values under different GUI densities, resolutions, or screen sizes have not been explored. Figure 2 shows that when s = 0.3, the desktop accuracy has decreased by 4%, indicating that the attention peaks at low resolutions are easily "overwhelmed" by large-sized controls, and the 8-neighborhood clustering easily merges controls with small spacings into the same area. The paper does not provide adaptive estimation of τ (such as Otsu, Pareto front) or online feedback strategies, resulting in the need for manual adjustment of hyperparameters, thereby weakening the "no-training" advantage.
>
> **[Response]**
>
> We thank the reviewer for the valuable suggestion on adaptive estimation of τ. Following the suggestion, we investigate the possibility of adaptive thresholding with three strategies—Otsu thresholding [r1] (**Otsu**), Pareto-front–style elbow heuristic [r2] (**Pareto front**), and entropy-based estimation [r3] (**Entropy**)—and evaluated them under both the low-resolution condition (s = 0.3) and the default setup (s = 0.5).
> For implementation, we directly apply each rule to the raw global attention map: Otsu is computed on a lightly smoothed attention map, the Pareto-front heuristic selects the largest drop in the sorted attention values, and the entropy-based method converts Shannon entropy into a threshold. These thresholds replace the fixed τ in the Global Pruning Stage, while all subsequent stages remain unchanged.
> | Model | Resize Ratio | Mobile Acc | Mobile TFLOPs | Web Acc | Web TFLOPs | Desktop Acc | Desktop TFLOPs | Avg Acc | Avg  TFLOPs |
> | --- | --- | --- | --- | --- | --- | --- | --- | --- | --- |
> | GUI-Actor-3B | X | 92.4 | 81.8 | 89.0 | 87.2 | 90.4 | 35.0 | 90.7 | 70.7 |
> | + Resizing | 0.3 | 89.0 | 5.2 | 74.1 | 5.6 | 44.0 | 2.8 | 72.1 | 4.7 |
> | + GOLD (τ = 0.12) | 0.3 | **91.8** | 9.2 | **85.4** | 9.7 | **85.6** | 9.7 | **88.0** | 8.8 |
> | + GOLD (*Pareto front*) | 0.3 | 88.4 | 9.2 | 74.1 | 7.2 | 63.5 | 4.2 | 77.0 | 6.3 |
> | + GOLD (*Entropy*) | 0.3 | 91.0 | 8.0 | 84.2 | 8.2 | 78.4 | 6.3 | 85.4 | 7.4 |
> | + GOLD (*Otsu*) | 0.3 | 91.0 | 8.8 | 85.8 | 9.0 | 84.4 | 5.6 | 87.5 | 8.2 |
> | + Resizing | 0.5 | 92.6 | 14.2 | 83.8 | 17.2 | 76.9 | 7.4 | 85.5 | 13.4 |
> | + GOLD (τ = 0.12) | 0.5 | **93.4** | 17.1 | **90.4** | 19.6 | 89.8 | 9.2 | **91.4** | 15.9 |
> | + GOLD (*Pareto front*) | 0.5 | 91.2 | 15.7 | 83.8 | 18.7 | 81.4 | 8.6 | 86.1 | 14.9 |
> | + GOLD (*Entropy*) | 0.5 | 93.2 | 16.5 | 88.8 | 19.2 | 88.0 | 9.2 | 90.3 | 15.4 |
> | + GOLD (*Otsu*) | 0.5 | 92.4 | 16.9 | 89.7 | 19.5 | 89.5 | 8.9 | 91.0 | 15.8 |
>
> The key findings are:
> - The fixed threshold (τ = 0.12) achieves the highest accuracy across all settings.
> - Among adaptive methods, Otsu shows the highest accuracy, followed by entropy-based estimation and Pareto-front selection.
> - While adaptive τ reduces the burden of manual hyperparameter selection, it does not fundamentally resolve the accuracy degradation at s = 0.3 on the Desktop split. This indicates that the degradation primarily arises from the loss due to low resolution rather than from the choice of τ.
> - Although the fixed threshold shows the strongest performance in our experiments, the results indicate clear potential for adaptive thresholding.
>
> We have incorporated the discussion, result, and details in our revised manuscript in `Appendix A.5.`
>
> [r1] Otsu, Nobuyuki. "A threshold selection method from gray-level histograms." Automatica 11.285-296 (1975): 23-27.
>
> [r2] Satopaa, Ville, et al. "Finding a" kneedle" in a haystack: Detecting knee points in system behavior." 2011 31st international conference on distributed computing systems workshops. IEEE, 2011.
>
> [r3] Shannon, Claude E. "A mathematical theory of communication." The Bell system technical journal 27.3 (1948): 379-423.

---

> ### Author Response · Authors · 2025-11-21
> **Author Response [2/4]**
>
> **[Weakness 2]**
> >When the instructions involve "selecting three checkboxes simultaneously" or "clicking the fifth item in the menu", the model needs to focus on ≥3 areas. GOLD first performs hard truncation and then fusion. Once the target control element does not enter the Top-C, it cannot be restored. The experiment only reports the success rate of single-click, does not calculate the Recall@C curve, and does not provide an ablation study for C > 3. It is difficult to prove that C = 3 is sufficient for complex tasks.
>
> **[Response]**
>
> We appreciate the reviewer’s concern that a fixed Top-C limit (C=3) might be insufficient for complex instructions. Our further analysis indicates that **C=3 already achieves the maximum accuracy** observed, and **larger C values offer no accuracy gain**.
> While we summarize the key findings here, we have added `Impact of crop count C` in `Section 4.4` for a complete discussion of this issue and included a visual example in `Figure 8` in Appendix.
>
> **(1) Recall@C: Accuracy saturates at C=3**
> We measure Recall@C, defined as the probability that the correct region (ground-truth) remains included among the Top-C candidate crops after the Global Pruning Stage. As expected, Recall@C increases slightly as C grows. However, despite this increase, the final grounding accuracy fully saturates at C=3, and **no further improvement** is observed for larger values of C.
>
> | C | Recall@C | Final Acc |
> | --- | --- | --- |
> | 1 | 95.36 | 91.27 |
> | 2 | 97.41 | 91.35 |
> | 3 | 97.80 | 91.43 |
> | 4 | 98.03 | 91.43 |
> | 5 | 98.19 | 91.43 |
>
> **(2) Rank distribution: the model effectively uses only rank ≤ 3**
> Without applying Top-C pruning, we tracked which regions are actually used by the Fusion Stage. **99.98% of all selected candidates come from Rank ≤ 3**, indicating that higher-rank regions play virtually no role in the final grounding.
>
> | Rank | Count | Percentage | Acc |
> | --- | --- | --- | --- |
> | 1 | 1257 | 98.82% | 90.78% |
> | 2 | 12 | 0.94% | 50.00% |
> | 3 | 2 | 0.16% | 100.00% |
> | 4 | 0 | 0.00% | – |
> | ≥5 | 1 | 0.08% | 0.00% |
>
> As shown, Rank-2 or Rank-3 candidates are occasionally selected after reversing the global score during Fusion, and lead to correct predictions. In contrast, regions with Rank ≥ 4 are never useful in our scenario: the only Rank-6 case was selected once but remained incorrect even after Fusion. Thus, increasing C beyond 3 has **no impact** on final accuracy. Therefore, **setting C ≥ 4 only increases FLOPs and peak memory without providing any practical benefit**, which is consistent with the above analysis.
>
> **(3) Concern about multi-element (ordinal) selection**
> For ordinal instructions such as “click the fifth item in the menu,” we find that the region corresponding to the **specified item consistently receives a sufficiently strong global attention response** to remain within the Top-3 candidate crops. In practice, the target item’s region is preserved during Global Pruning, and the Fusion step selects it based on the stronger global signal. As shown in `Figure 8` in the Appendix, such ordinal elements are reliably retained through pruning and correctly grounded.
>
> **(4) Concern about multi-click instructions**
> In multi-click environments (e.g., Mind2Web [r4]), the LLM Planner decomposes a multi-element instruction into several single-click grounding steps. Multi-element reasoning is handled by the planner, whereas the grounding model performs **single-element** grounding. Under this structure, GOLD has no architectural disadvantage compared to the vanilla model.
>
> [r4] Deng, Xiang, et al. "Mind2web: Towards a generalist agent for the web." Advances in Neural Information Processing Systems 36 (2023): 28091-28114.

---

> ### Author Response · Authors · 2025-11-21
> **Author Response [3/4]**
>
> **[Weakness 3]**
> >The author assumes that "the low-resolution attention peak is sufficient to indicate the high-resolution target location", but when image compression causes text blurring and icons to have jagged edges, VLM attention often disperses to adjacent text or blank areas. Figure 4 is an example that fails due to blurring, but the paper only qualitatively presents this, without quantifying the failure rate, nor providing secondary verification (such as edge gradients, OCR boxes) to correct the candidate regions. Once the attention map is unreliable, the three-step pipeline of GOLD will go completely wrong, lacking rollback or self-check mechanisms.
>
> **[Response]**
>
> Sorry for the confusion. The reviewer’s concern is based on the assumption that GOLD relies on the **exact location of a single low-resolution attention peak** to determine the final grounding point. This is **not** how GOLD operates. The Global Stage does *not* use the peak position directly; instead, what matters is whether the **ground-truth region is included among the Top-C candidate crops** extracted from the low-resolution attention map.
>
> `Figure 4` may appear to show a shifted or blurred peak, but **this does not imply that GOLD fails**. Even when the peak is imperfect, the correct region is still included among the top-ranked clusters in most cases. What GOLD requires is **region-level recall**, not pixel-level precision. We revised `Figure 4` and its description in `Appendix A.3` to avoid confusion.
>
> During the rebuttal period, we have quantitatively verified this by the Recall@C evaluation presented in `Section 4.4` and `Table 5`:
> **97.8% of ground-truth targets appear within the Top-3 regions**, meaning that the Local Refine Stage almost always receives a high-resolution crop that contains the correct target. The remaining 2.2% constitutes the true failure rate of the Global Stage—not the far larger number implied by blurred peaks.
>
> Therefore, the reviewer’s statement that “a low-resolution peak is insufficient, causing the entire pipeline to fail” does not reflect GOLD’s design. GOLD tolerates blurred or slightly shifted peaks because it extracts *regions*, not points. As long as the correct region remains in the Top-C (which occurs 97.8% of the time), the subsequent Local and Fusion stages operate correctly.
>
> The reviewer’s suggestion to incorporate OCR boxes or edge-gradient verification is meaningful for high-reliability settings. However, GOLD is intentionally designed as a **lightweight, zero-training, low-overhead pipeline**, and such mechanisms are better suited as optional extensions rather than core components. Within the scope of this work, the region-level recall of the Global Stage is already high, and GOLD performs robustly without additional rollback or self-checking procedures.

---

> ### Author Response · Authors · 2025-11-21
> **Author Response [4/4]**
>
> **[Weakness 4]**
> >To reduce the number of forward passes in VLM, the authors combined C high-resolution cropped images into multiple images for input at once. Although this approach is efficient, the self-attention mechanism of VLM only computes within a single image, and there is no interaction between regions. For instructions like "Click the 'Save' button in Panel A instead of the 'Save' button in Panel B", which require comparing two similar controls, the model cannot utilize cross-region attention for disambiguation. It can only rely on subsequent score fusion, but the fusion stage lacks an inter-region comparison mechanism.
>
> **[Response]**
>
> We appreciate the reviewer’s thoughtful observation regarding the limitation of combining multiple high-resolution crops into a single VLM input. While we agree that this design might prevent explicit cross-region self-attention, this does not hinder GOLD’s performance in practice, as region-level disambiguation is handled earlier in the pipeline.
>
> Specifically, we clarify that GOLD does not rely fully on the Local Refinement Stage to compare different regions; the primary role of the Local Refinement Stage is to refine the click location within each candidate region. Region-level comparison—such as distinguishing “the Save button in Panel A” from “the Save button in Panel B”—is resolved by the Global Pruning Stage, which evaluates attention over the entire screen. Finally, the Fusion Stage combines the global region scores with the local evidence, providing effective cross-region disambiguation without requiring explicit self-attention across crops. `Figure 8` in the Appendix shows a scenario similar to the reviewer’s example, where two “subscribe” buttons appear nearly identical without contextual information. In this case, the correct region receives a stronger global signal during the Global Pruning Stage, allowing GOLD to retain the correct crop and ultimately select the appropriate button during Fusion.
>
> We agree with the reviewer’s insight that explicit cross-region reasoning may be beneficial in cases involving highly symmetric or visually identical components. While GOLD is designed with efficiency as its primary objective, we acknowledge that richer inter-region interaction could further improve accuracy in rare edge cases. Since GOLD is, to our knowledge, the first framework to perform multi-crop high-resolution grounding within a cost-efficient Global–Local–Fusion architecture, adding more advanced cross-region mechanisms would introduce an accuracy-efficiency tradeoff that warrants careful consideration. We view this as a valuable direction for future work and have included a discussion of this point in `Section 5`.

---

### Official Review · Reviewer_EAVr · 2025-11-01

**Soundness:** 2
**Presentation:** 3
**Contribution:** 2
**Rating:** 4
**Confidence:** 3

**Summary:**

This paper aims to address the computationally burdensome nature of GUI grounding in high-resolution scenarios. It proposes GOLD, a tuning-free approach that enhances grounding performance by combining global pruning with local refinement and coupling their predictions. Experiments demonstrate that the method substantially reduces TFLOPs while achieving improved performance.

**Strengths:**

1. The overall presentation is clear.

2. The final experimental evaluation is fairly comprehensive.

**Weaknesses:**

1. The motivation of the method is somewhat unclear. For example, Section 3.3 argues that local refinement recovers details that may be lost at low resolution. However, in Figure 1, the local refinement appears to be inaccurate, and the final prediction relies on fusion via α. From Figure 3, it also seems that performance is still primarily driven by the global view. Therefore, the motivation for local refinement feels weak.

2. The method’s novelty appears limited. For instance, Line 196 notes that the proposed approach borrows ideas from prior work.

**Questions:**

1. In Section 3.2, which layer’s attention map is used? How would using attention maps from different layers affect the results?

---

> ### Author Response · Authors · 2025-11-21
> **Author Response**
>
> We sincerely appreciate your time and effort in providing these valuable comments. We address your concerns and questions below, and we have also uploaded an `updated manuscript` that incorporates our rebuttal.
>
> ---
>
> **[Weakness 1]**
> >The motivation of the method is somewhat unclear. For example, Section 3.3 argues that local refinement recovers details that may be lost at low resolution. However, in Figure 1, the local refinement appears to be inaccurate, and the final prediction relies on fusion via α. From Figure 3, it also seems that performance is still primarily driven by the global view. Therefore, the motivation for local refinement feels weak.
>
>
> **[Response]**
>
> Thank you for the comment. Figure 1 shows only one case, and many other examples better illustrate why local refinement is needed. As shown in `Figures 6(a)` and `6(b)`, the global view often cannot resolve small elements such as the “add tab” button, which motivates adding a high-resolution refinement step.
>
> The importance of the local refinement stage is also supported quantitatively. In `Figure 2(c)`, **using only the global stage (i.e., GUI-Actor-3B) leads to about a 40% accuracy drop** (85.6%→44.0%) at s=30%, indicating that downsampled inputs lose essential details. These results show that while the global stage provides useful coarse localization, local refinement is required for accurate grounding.
>
> In addition, we clarify that the refinement module is intended to operate within the globally chosen region rather than serve as a standalone predictor. It is not meant to decide which crop is correct, but to localize precisely inside the region selected by the global stage. Thus, the behavior in Figure 1 does not contradict our motivation, as refinement is designed to complement—not replace—the global stage.
>
> ---
>
> **[Weakness 2]**
> >The method’s novelty appears limited. For instance, Line 196 notes that the proposed approach borrows ideas from prior work.
>
> **[Response]**
>
> Sorry for the confusion. Our method and contributions are distinct from Zhang et al. [r1] in Line 196. We have revised the text in  `Section 3.2` to avoid any potential misunderstanding as follows:
> > While recent GUI grounding work has explored using attention maps for region selection (Wu et al. (2025); Zhang et al. (2025); Xu et al. (2025)), our contribution lies in making this process cost-efficient through an effective pruning stage followed by local refinement and fusion stages.
>
> Specifically, using attention maps to identify vision tokens relevant to each text token is a widely used technique, and Zhang et al. is simply a representative example of this general practice. However, general localization methods such as Zhang et al. primarily aim to improve performance on high-resolution images and often increase computational cost—for example, by directly concatenating a large number of tokens—accepting this overhead to maximize task accuracy.
> In contrast, our goal is to reduce computation cost. To achieve this, we apply **downsampling** and then **mitigate its side effects** (e.g., drastic grounding accuracy drop as shown in `Figure 2 (c))` by fusing global and local context for the final grounding decision. This global-local fusion strategy allows us to maintain accuracy while significantly lowering the computational burden.
>
> [r1] Zhang, Jiarui, et al. "MLLMs Know Where to Look: Training-free Perception of Small Visual Details with Multimodal LLMs." ICLR 2025.
>
>
> ---
>
> **[Question 1]**
> >In Section 3.2, which layer’s attention map is used? How would using attention maps from different layers affect the results?
>
> **[Response]**
>
> Our method uses GUI-Actor as the main backbone model, and **it does not operate based on naive attention-map inspection [r2]**; instead, it employs a separate action head that explicitly attends to relevant vision tokens. Therefore, the issue of attention-map layer selection doesn’t apply to GUI-Actor; it only applies to Qwen 2.5 VL. For Qwen 2.5 VL, we chose deeper layers (20–31) based on the finding in [r3] that deeper layers, compared to shallower ones, place greater attention on vision tokens associated with instruction tokens. We also conducted an ablation study on layer choices and have added the results to `Table 8` in `Appendix A.4`.
>
>
>
>
> [r2] Wu, Qianhui, et al. "GUI-Actor: Coordinate-Free Visual Grounding for GUI Agents." arXiv preprint arXiv:2506.03143 (2025).
>
> [r3] Luo, Junwei, et al. "When Large Vision-Language Model Meets Large Remote Sensing Imagery: Coarse-to-Fine Text-Guided Token Pruning." ICCV 2025.

---

### Author Response · Authors · 2025-11-27
**Rebuttal Summary and a Gentle Reminder**

Dear Reviewers,

We appreciate all of you for your positive reviews and for highlighting the **strengths of our work**:

**EAVr**
- Clear overall presentation
- Comprehensive experimental evaluation.

**f7o6**
- Training-free, plug-and-play, and compatible with existing VLMs
- A well-structured three-stage pipeline balancing global context and local detail
- Strong acceleration while improving accuracy across multiple benchmarks with thorough experiments
- Robustness in dense and foreground-heavy GUIs compared to prior pruning and token-reduction methods.

**FMFj**
- Practical and seamlessly deployable without fine-tuning
- Lightweight and intuitive global-local-fusion design
- Strong empirical performance with significant TFLOP reduction and consistent gains across backbones and platforms supported by extensive ablations
- Clear writing, strong figures, and reproducible methodology.


We also sincerely thank the reviewers for their **constructive comments** to improve our work. We have **addressed all the questions from reviewers** with clarifications and new experiments during this rebuttal period. We summarize how we addressed the reviewers’ main questions as follows:



**EAVr**

- We clarified the motivation behind our proposed method using `Figures 6(a)`, `6(b)`, and `2(c)`, which provide qualitative and quantitative evidence supporting the need for local refinement.
- We clarified our novelty and revised the text to distinguish our contribution from prior work. We have revised the text in `Section 3.2` accordingly.
- We explained which attention layers are used and added an ablation experiment on layer choices in `Table 8` of `Appendix A.4`.



**f7o6**

- We explored adaptive thresholding strategies (Otsu, Pareto front, entropy-based) and added results in `Table 10` of `Appendix A.5`.
- We analyzed the effect of Top-C and demonstrated that performance saturates at C=3, adding Recall@C and related discussion in `Section 4.4`.
- We demonstrated through quantitative results that the method reliably selects regions containing ground truth, even when the  screenshots are blurry. We added the corresponding results to `Table 5` in `Section 4.4`.
- We clarified that cross-region disambiguation is handled by the Global and Fusion stages and included this clarification and discussion of potential extensions in `Section 5`.



**FMFj**

- We clarified how GOLD differs from GUI-Actor and highlighted the efficiency benefits of the global-to-local pipeline. We have also revised the text in `Section 3.2` accordingly.
- We broadened our comparisons with SparseVLM and FastV by adding updated results and more extensive hyperparameter sweeps, which are presented in `Table 3` and `Table 9`.
- We provided end-to-end latency measurements showing that TFLOP savings translate to real runtime speedup. We have included this result in `Appendix A.6` and `Table 11`.
- We clarified task scope, GUI-specific design choices, and applicability beyond GUI benchmarks. We have revised `Section 3.4` accordingly.
- We demonstrated generalizability by applying GOLD to additional models such as ZonUI. We have incorporated the updated results into `Table 3`.


Also, we’ve uploaded an updated manuscript that incorporates our rebuttal, with changes highlighted in $\color{blue}{\text{\textit{blue}}}$.
As the discussion period ends in about a week, we remain fully available for any further questions or clarifications and are happy to provide additional explanations as needed.

Best regards,

Authors

---

### Meta-Review · Area_Chair_ZiCn · 2026-01-06

**Summary:**

This submission proposes GOLD, a training-free global-to-local pipeline to reduce the cost of VLM-based GUI grounding by (i) running a downsampled global pass to identify candidate regions via attention, (ii) running a high-resolution local refinement only on cropped regions, and (iii) performing global–local fusion without additional VLM passes. The paper reports large efficiency gains (e.g., ~78% TFLOP reduction when integrated with GUI-Actor on ScreenSpot-V2) while maintaining or slightly improving accuracy, and provides additional evaluations/ablations and latency numbers in the revision.

Across the three reviews, there is broad agreement that the work is practical, clear, and empirically strong. The initial scores were mixed. The main concerns driving borderline/reject scores were: role of the local refinement stage, how much is fundamentally new beyond standard coarse-to-fine grounding and prior attention-based region selection, hyperparameter sensitivity, baselines and practicality of efficiency claims, cross-region disambiguation, and generalization beyond GUIs.

In the rebuttal, the authors add experiments and clarifications (e.g., Recall@C and crop-rank analysis for C, adaptive τ ablations, additional pruning baselines and FastV tuning, attention-layer ablations for Qwen2.5‑VL, and latency numbers). Many of the implementation and evaluation completeness concerns are addressed in the revised manuscript. The remaining point that most strongly affects the final decision is novelty: even with clearer writing, the method is still a relatively direct coarse-to-fine application to GUI grounding. That said, the work’s practical impact on speedups is significant.

**Reviewer Concerns:**

**Reviewer EAVr**

Concerns addressed

Motivation for local refinement: The rebuttal clarifies that refinement is not intended to pick the correct region among crops but to refine within globally selected regions, and provides qualitative/quantitative evidence that downsampling alone can cause major accuracy loss that refinement mitigates.

Which attention layer is used: The authors clarify that GUI-Actor uses an action head rather than naive layer attention selection, and they add an explicit layer-range ablation for Qwen2.5‑VL in the appendix

Concerns still outstanding

Novelty: While the rebuttal improves the explanation and distinguishes the goal (cost-efficient pruning + refinement + fusion) from prior “look where to look” approaches, the underlying mechanism remains a coarse-to-fine recipe with attention-guided region selection. This is partially mitigated by the strong empiricalvalue, but the novelty concern is not fully eliminated.

**Reviewer f7o6**

Concerns addressed

Fixed τ and lack of adaptive estimation: The authors implement and evaluate adaptive thresholding strategies, showing that the fixed τ works best among tested options while adaptive methods are viable drop-ins.

Crop count ablation: The authors add Recall@C and show final accuracy saturates at C=3, and provide analysis of which crop ranks are actually used in fusion.

Unreliability of low-res attention peaks: The rebuttal clarifies the method relies on region-level recall rather than exact peak location, and supports this with the Recall@C numbers.

Cross-region disambiguation without crop-level interaction: The authors explain that global-stage full-screen attention provides cross-region context and fusion re-weights local candidates accordingly.

Concerns still outstanding

Robustness beyond the analyzed settings: The critique about failure modes under severe downsampling is not fully resolved.

Hard cases requiring richer multi-region reasoning: The paper now acknowledges the potential need for explicit cross-region mechanisms in symmetric/duplicate-control scenarios.

**Reviewer FMFj**

Concerns addressed

Broader efficiency-oriented comparisons: The authors expand comparisons with training-free acceleration/pruning baselines.

Latency: The authors add end-to-end latency measurements showing significant wall-clock speedups.

Clarifying difference from GUI-Actor: The rebuttal explicitly contrasts GUI-Actor’s full-resolution processing with GOLD’s downsampled pruning + targeted refinement + fusion, and updates text accordingly.

Concerns still outstanding

Coarse-to-fine paradigm: Even with clearer positioning, the method remains closely aligned with established coarse-to-fine ideas and attention-based candidate selection; the paper’s value is primarily practical efficiency, not conceptual novelty.

Breadth beyond GUI grounding: The authors argue GUIs are the intended scope and explain why RefCOCO-style datasets may not highlight the same hierarchical properties.

**Reviewer Scores:**

Reviewer EAVr: 4 → 4, possible of 6

The layer-selection question is answered. The motivation for local refinement is clarified. Remaining novelty concerns likely prevent a jump to 6.

Reviewer f7o6: 4 → 6, possible of 4

The rebuttal adds exactly the analyses requested: adaptive τ ablations, Recall@C and C>3 behavior, and rank-distribution evidence, plus clarifications on blur tolerance and disambiguation.

Reviewer FMFj: 6 → 6

The main actionable requests (broader baselines, FastV tuning clarity, SparseVLM comparison, and latency) are addressed in the revision. However, the reviewer’s remaining reservations about novelty and scope beyond GUIs are not fundamentally resolved.

---

### Decision · Program_Chairs · 2026-01-26

Reject